# Federated Online Prediction from Experts with Differential Privacy: Separations and Regret Speed-ups

**Fengyu Gao,   Ruiquan Huang,   Jing Yang**
School of EECS, The Pennsylvania State University, USA
{fengyugao, rzh5514, yangjing}@psu.edu

## Abstract

We study the problems of differentially private federated online prediction from experts against both *stochastic adversaries* and *oblivious adversaries*. We aim to minimize the average regret on $m$ clients working in parallel over time horizon $T$ with explicit differential privacy (DP) guarantees. With stochastic adversaries, we propose a **Fed-DP-OPE-Stoch** algorithm that achieves $\sqrt{m}$-fold speed-up of the per-client regret compared to the single-player counterparts under both pure DP and approximate DP constraints, while maintaining logarithmic communication costs. With oblivious adversaries, we establish non-trivial lower bounds indicating that *collaboration among clients does not lead to regret speed-up with general oblivious adversaries*. We then consider a special case of the oblivious adversaries setting, where there exists a low-loss expert. We design a new algorithm **Fed-SVT** and show that it achieves an $m$-fold regret speed-up under both pure DP and approximate DP constraints over the single-player counterparts. Our lower bound indicates that Fed-SVT is nearly optimal up to logarithmic factors. Experiments demonstrate the effectiveness of our proposed algorithms. To the best of our knowledge, this is the first work examining the differentially private online prediction from experts in the federated setting.

## 1   Introduction

Federated Learning (FL) (McMahan et al., 2017) is a distributed machine learning framework, where numerous clients collaboratively train a model by exchanging model update through a server. Owing to its advantage in protecting the privacy of local data and reducing communication overheads, FL is gaining increased attention in the research community, particularly in the online learning framework (Mitra et al., 2021; Park et al., 2022; Kwon et al., 2023; Gauthier et al., 2023). Noticeable advancements include various algorithms in federated multi-armed bandits (Shi et al., 2021; Huang et al., 2021; Li and Wang, 2022; Yi and Vojnovic, 2022, 2023), federated online convex optimization (Patel et al., 2023; Kwon et al., 2023; Gauthier et al., 2023), etc.

Meanwhile, differential privacy (DP) has been integrated into online learning, pioneered by Dwork et al. (2010). Recently, Asi et al. (2022b) studied different types of adversaries, developing some of the best existing algorithms and establishing lower bounds. Within the federated framework, although differentially private algorithms have been proposed for stochastic bandits (Li et al., 2020b; Zhu et al., 2021; Dubey and Pentland, 2022) and linear contextual bandits (Dubey and Pentland, 2020; Li et al., 2022a; Zhou and Chowdhury, 2023; Huang et al., 2023), to the best of our knowledge, federated online learning in the adversarial setting with DP considerations remain largely unexplored.

38th Conference on Neural Information Processing Systems (NeurIPS 2024).

In this work, we focus on federated online prediction from experts (OPE) with rigorous differential privacy (DP) guarantees[1]. OPE (Arora et al., 2012) is a classical online learning problem under which, a player chooses one out of a set of experts at each time slot and an adversary chooses a loss function. The player incurs a loss based on its choice and observes the loss function. With all previous observations, the player needs to decide which expert to select each time to minimize the cumulative expected loss. We consider two types of adversaries in the context of OPE. The first type, *stochastic adversary*, chooses a distribution over loss functions and samples a loss function independently and identically distributed (IID) from this distribution at each time step. The second type, *oblivious adversary*, chooses a sequence of loss functions in advance. [2]

When extending the OPE problem to the federated setting, we assume that the system consists of a central server and $m$ local clients, where each client chooses from $d$ experts to face an adversary at each time step over time horizon $T$. The server coordinates the behavior of the clients by aggregating the clients' updates to form a global update (predicting a new global expert), while the clients use the global expert prediction to update its local expert selection and compute local updates. The local updates will be sent to the server periodically. In the Federated OPE framework, clients face either *stochastic adversaries*, receiving loss functions from the same distribution in an IID fashion, or *oblivious adversaries*, which arbitrarily select loss functions for each client at each time step beforehand (Yi and Vojnovic, 2023). Specifically, we aim to answer the following question:

*Can we design differentially private federated OPE algorithms to achieve regret speed-up against both stochastic and oblivious adversaries?*

In this paper, we give definitive answers to the question. Our contributions are summarized as follows.

- **Speed-up for stochastic adversaries.** We develop a communication-efficient algorithm Fed-DP-OPE-Stoch for stochastic adversaries with DP guarantees. The algorithm features the following elements in its design: 1) *Local loss function gradient estimation for global expert determination.* To reduce communication cost, we propose to estimate the gradient of each client's previous loss functions locally, and only communicate these estimates to the server instead of all previous loss functions. 2) *Local privatization process.* Motivated by the need for private communication in FL, we add noise to client communications (local gradient estimates) adhering to the DP principles, thereby building differentially private algorithms.

  We show that Fed-DP-OPE-Stoch achieves $1/\sqrt{m}$-fold reduction of the per-client regret compared to the single-player counterparts (Asi et al., 2022b) under both pure DP and approximate DP constraints, while maintaining logarithmic communication costs.

- **Lower bounds for oblivious adversaries.** We establish new lower bounds for federated OPE with oblivious adversaries. Our findings reveal a critical insight: collaboration among clients does not lead to regret speed-up in this context. Moreover, these lower bounds highlight a separation between oblivious adversaries and stochastic adversaries, as the latter is necessary to reap the benefits of collaboration in this framework.

  Formulating an instance of oblivious adversaries for the corresponding lower bounds is a non-trivial challenge, because it requires envisioning a scenario where the collaborative nature of FL does not lead to the expected improvements in regret minimization. To deal with this challenge, we propose a *policy reduction approach in FL*. By defining an "average policy" among all clients against a uniform loss function generated by an oblivious adversary, we reduce the federated problem to a single-player one, highlighting the equivalence of per-client and single-player regret. Our lower bounds represent the first of their kind for differentially private federated OPE problems.

- **Speed-up for oblivious adversaries under realizability assumption.** We design a new algorithm Fed-SVT that obtains near-optimal regret when there is a low-loss expert. We show that Fed-SVT achieves an $m$-fold speed-up in per-client regret when compared to single-player models (Asi et al., 2023). This shows a clear separation from the general setting where collaboration among clients does not yield benefits in regret reduction when facing oblivious adversaries. Furthermore, we establish a new lower bound in the realizable case. This underlines that our upper bound is nearly optimal up to logarithmic factors.

---

[1] We provide some applications of differentially private federated OPE in Appendix B.

[2] We don't consider adaptive adversaries in this work because they can force DP algorithms to incur a linear regret, specifically when $\varepsilon \leq 1/10$ for $\varepsilon$-DP (Theorem 10 in Asi et al. (2022b)).

Table 1: Comparisons for Online Prediction from Experts under DP Constraints

| Adversaries | Algorithm (Reference) | Model | DP | Regret | Communication cost |
|---|---|---|---|---|---|
| Stochastic | Limited Updates (Asi et al., 2022b) | SING | $\varepsilon$-DP, $(\varepsilon, \delta)$-DP | $O\left(\sqrt{T \log d} + \frac{\log d \log T}{\varepsilon}\right)$ | - |
| | Fed-DP-OPE-Stoch (**Corollary 1**) | FED | $\varepsilon$-DP, $(\varepsilon, \delta)$-DP | $O\left(\sqrt{\frac{T \log d}{m}} + \frac{\log d \log T}{\sqrt{m}\varepsilon}\right)$ | $O\left(md \log T\right)$ |
| Oblivious (Realizable) | Sparse-Vector (Asi et al., 2023) | SING | $\varepsilon$-DP | $O\left(\frac{\log T \log d + \log^2 d}{\varepsilon}\right)$ | - |
| | Fed-SVT (**Theorem 4**) | FED | $\varepsilon$-DP | $O\left(\frac{\log T \log d + \log^2 d}{m\varepsilon} + N \log d\right)$ | $O\left(mdT/N\right)$ |
| | Sparse-Vector (Asi et al., 2023) | SING | $(\varepsilon, \delta)$-DP | $O\left(\frac{\log T \log d + \log^{3/2} d}{\varepsilon}\right)$ | - |
| | Fed-SVT (**Theorem 4**) | FED | $(\varepsilon, \delta)$-DP | $O\left(\frac{\log T \log d + \log^{3/2} d}{m\varepsilon} + N \log d\right)$ | $O\left(mdT/N\right)$ |
| | Lower bound (**Theorem 3**) | FED | $\varepsilon$-DP, $(\varepsilon, \delta)$-DP | $\Omega\left(\frac{\log d}{m\varepsilon}\right)$ | - |
| Oblivious | Lower bound (**Theorem 2**) | FED | $\varepsilon$-DP, $(\varepsilon, \delta)$-DP | $\Omega\left(\min\left(\frac{\log d}{\varepsilon}, T\right)\right)$ | - |

$m$: number of clients; $T$: time horizon; $d$: number of experts; $\varepsilon, \delta$: DP parameters; SING and FED stand for single-client and federated settings, respectively.

A summary of our main results and how they compare with the state of the art is shown in Table 1. A comprehensive literature review is provided in Appendix A.

## 2 Preliminaries

**Federated online prediction from experts.** Federated OPE consists of a central server, $m$ clients and an interactive $T$-round game between an adversary and an algorithm. At time step $t$, each client $i \in [m]$ first selects an expert $x_{i,t} \in [d]$, and then, the adversary releases a loss function $l_{i,t}$. *Stochastic adversaries* choose a distribution over loss functions and sample a sequence of loss functions $l_{1,1}, \ldots, l_{m,T}$ in an IID fashion from this distribution, while *oblivious adversaries* choose a sequence of loss functions $l_{1,1}, \ldots, l_{m,T}$ at the beginning of the game.

For federated OPE, the utility of primary interest is the expected cumulative regret among all clients defined as:

$$\text{Reg}(T, m) = \frac{1}{m}\left[\sum_{i=1}^{m}\sum_{t=1}^{T} l_{i,t}(x_{i,t}) - \min_{x^\star \in [d]} \sum_{i=1}^{m}\sum_{t=1}^{T} l_{i,t}(x^\star)\right].$$

**Differential privacy.** We define differential privacy in the online setting following (Dwork et al., 2010). If an adversary chooses a loss sequence $\mathcal{S} = (l_{1,1}, \ldots, l_{m,T})$, we denote $\mathcal{A}(\mathcal{S}) = (x_{1,1}, \ldots, x_{m,T})$ the output of the interaction between the federated online algorithm $\mathcal{A}$ and the adversary. We say $\mathcal{S} = (l_{1,1}, \ldots, l_{m,T})$ and $\mathcal{S}' = (l'_{1,1}, \ldots, l'_{m,T})$ are neighboring datasets if $\mathcal{S}$ and $\mathcal{S}'$ differ in a one element. [3].

**Definition 1** (($\varepsilon, \delta$)-DP). A randomized federated algorithm $\mathcal{A}$ is $(\varepsilon, \delta)$-differentially private against an adversary if, for all neighboring datasets $\mathcal{S}$ and $\mathcal{S}'$ and for all events $\mathcal{O}$ in the output space of $\mathcal{A}$, we have

$$\mathbb{P}[\mathcal{A}(\mathcal{S}) \in \mathcal{O}] \leq e^\varepsilon \mathbb{P}\left[\mathcal{A}\left(\mathcal{S}'\right) \in \mathcal{O}\right] + \delta.$$

**Communication model.** Our setup involves a central server facilitating periodic communication with zero latency with all clients. Specifically, the clients can send "local model updates" to the central server, which then aggregates and broadcasts the updated "global model" to the clients. We assume full synchronization between clients and the server (McMahan et al., 2017). Following Wang et al. (2019), we define the communication cost of an algorithm as the number of scalars (integers or real numbers) communicated between the server and clients.

## 3 Federated OPE with Stochastic Adversaries

In this section, we aim to design an algorithm for Fed-DP-OPE with stochastic adversaries that achieves regret speed-up compared to the single-player setting under DP constraints with low

---

[3]We note that our definition is slightly different from that in Asi et al. (2022b), where the loss functions are explicitly dependent on a sequence of variables $z_i, \ldots, z_T \in \mathcal{Z}$. This is because we focus on stochastic and oblivious adversaries in this work, and our definition is essentially equivalent to that in Asi et al. (2022b) for those cases.

communication cost. We consider the loss functions $l_{1,1}(\cdot), \ldots, l_{m,T}(\cdot)$ to be convex, $\alpha$-Lipschitz and $\beta$-smooth w.r.t. $\| \cdot \|_1$ in this section.

## 3.1 Intuition behind Algorithm Design

To gain a better understanding of our algorithm design, we first elaborate the difficulties encountered when extending prevalent OPE models to the FL setting under DP constraints. It is worth noting that all current OPE models with stochastic adversaries rely on gradient-based optimization methods. The central task in designing OPE models with stochastic adversaries lies in leveraging past loss functions to guide the generation of expert predictions. Specifically, we focus on the prominent challenges associated with the widely adopted Frank-Wolfe-based methods (Asi et al., 2022b). This algorithm iteratively moves the expert selection $x_t$ towards a point that minimizes the gradient estimate derived from the past loss functions $l_1, \ldots, l_{t-1}$ over the decision space $\mathcal{X}$, where $\mathcal{X} = \Delta_d = \left\{ x \in \mathbb{R}^d : x_i \geq 0, \sum_{i=1}^{d} x_i = 1 \right\}$, and each $x \in \mathcal{X}$ represents a probability distribution over $d$ experts. With DP constraints, a tree-based method is used for private aggregation of the gradients of loss functions (Asi et al., 2021b).

In the federated setting, it is infeasible for the central server to have full access to past loss functions due to the high communication cost. To overcome this, we use a design of *local loss function gradient estimation for global expert determination*. Our solution involves locally estimating the gradient of each client's loss functions, and then communicating these estimates to the server, which globally generates a new prediction. This strategy bypasses the need for full access to all loss functions, reducing the communication overhead while maintaining efficient expert selection.

To enhance privacy in the federated system, we implement a *local privatization process*. When communication is triggered, clients send "local estimates" of the gradients of their loss functions to the server. These local estimates include strategically added noise, adhering to DP principles. The privatization method is crucial as it extends beyond privatizing the selection of experts; it ensures the privacy of all information exchanged between the central server and clients within the FL framework.

## 3.2 Algorithm Design

To address the aforementioned challenges, we propose the Fed-DP-OPE-Stoch algorithm. The Fed-DP-OPE-Stoch algorithm works in phases. In total, it has $P$ phases, and each phase $p \in [P]$ contains $2^{p-1}$ time indices. Fed-DP-OPE-Stoch contains a client-side subroutine (Algorithm 1) and a server-side subroutine (Algorithm 2). The framework of our algorithm is outlined as follows.

At the initialization phase, the server selects an arbitrary point $z \in \mathcal{X}$ and broadcasts to all clients. Subsequently, each client initializes its expert selection $x_{i,1} = z$, pays cost $l_{i,1}(x_{i,1})$ and observes the loss function.

Starting at phase $p = 2$, each client uses loss functions from the last phase to update its local loss function gradient estimation, and then coordinates with the server to update its expert selection. After that, it sticks with its decision throughout the current phase and observes the loss functions. We elaborate this procedure as follows.

**Private Local Loss Function Gradient Estimation.** At the beginning of phase $p$, each client privately estimates the gradient using local loss functions from the last phase $\mathcal{B}_{i,p} = \{l_{i,2^{p-2}}, \ldots, l_{i,2^{p-1}-1}\}$. We employ the tree mechanism for the private aggregation at each client, as in the DP-FW algorithm from Asi et al. (2021b). Roughly speaking, DP-FW involves constructing binary trees and allocating sets of loss functions to each vertex. The gradient at each vertex is then estimated using the loss functions of that vertex and the gradients along the path to the root. Specifically, we run a DP-FW subroutine at each client $i$, with sample set $\mathcal{B}_{i,p}$, parameter $T_1$ and batch size $b$. In DP-FW, each vertex $s$ in the binary tree $j$ corresponds to a gradient estimate $v_{i,j,s}$. DP-FW iteratively updates gradient estimates by visiting the vertices of binary trees. The details of DP-FW can be found in Algorithm 3 in Appendix C.1.

Intuitively, in the DP-FW subroutine, reaching a leaf vertex marks the completion of gradient refinement for a specific tree path. At these critical points, we initiate communication between the central server and individual clients. Specifically, as the DP-FW subroutine reaches a leaf vertex $s$ of tree $j$, each client sends the server a set of noisy inner products for each decision set vertex

$c_n$ with their gradient estimate $v_{i,j,s}$. In other words, each client communicates with the server by sending $\{\langle c_n, v_{i,j,s}\rangle + \xi_{i,n}\}_{n\in[d]}$, where $c_1, \ldots, c_d$ represents $d$ vertices of decision set $\mathcal{X} = \Delta_d$, $\xi_{i,n} \sim \text{Lap}(\lambda_{i,j,s})$, and $\lambda_{i,j,s} = \frac{4\alpha 2^j}{b\varepsilon}$.

**Private Global Expert Prediction.** After receiving $\{\langle c_n, v_{i,j,s}\rangle + \xi_{i,n}\}_{n\in[d]}$ from all clients, the central server privately predicts a new expert:

$$\bar{w}_{j,s} = \underset{c_n:1\leq n\leq d}{\arg\min} \left[ \frac{1}{m}\sum_{i=1}^{m}\left(\langle c_n, v_{i,j,s}\rangle + \xi_{i,n}\right) \right]. \tag{1}$$

Subsequently, the server broadcasts the "global prediction" $\bar{w}_{j,s}$ to all clients.

**Local Expert Selection Updating.** Denote the index of the leaf $s$ of tree $j$ as $k$. Then, upon receiving the global expert selection $\bar{w}_{j,s}$, each client $i$ updates its expert prediction for leaf $k+1$, denoted as $x_{i,p,k+1} \in \mathcal{X}$, as follows:

$$x_{i,p,k+1} = (1 - \eta_{i,p,k})x_{i,p,k} + \eta_{i,p,k}\bar{w}_{j,s}, \tag{2}$$

where $\eta_{i,p,k} = \frac{2}{k+1}$.

After updating all leaf vertices of the trees, the client obtains $x_{i,p,K}$, the final state of the expert prediction in phase $p$. Then, each client $i$ sticks with expert selection $x_{i,p,K}$ throughout phase $p$ and collects loss functions $l_{i,2^{p-1}}, \ldots, l_{i,2^p-1}$.

*Remark* 1 (Comparison with Asi et al. (2022b)). The key difference between Fed-DP-OPE-Stoch and non-federated algorithms (Asi et al., 2022b) lies in our innovative approach to centralized coordination and communication efficiency. Unlike the direct application of DP-FW in each phase in Asi et al. (2022b), our algorithm employs *local loss function gradient estimation*, *global expert prediction* and *local expert selection updating*. Additionally, our strategic communication protocol, where clients communicate with the server only when DP-FW subroutine reaches leaf vertices, significantly reduces communication costs. Moreover, the integration of DP at the local level in our algorithm distinguishes it from non-federated approaches.

---

**Algorithm 1** Fed-DP-OPE-Stoch: Client $i$

---

1: **Input:** Phases $P$, trees $T_1$, decision set $\mathcal{X} = \Delta_d$ with vertices $\{c_1, \ldots, c_d\}$, batch size $b$.
2: **Initialize:** Set $x_{i,1} = z \in \mathcal{X}$ and pay cost $l_{i,1}(x_{i,1})$.
3: **for** $p = 2$ to $P$ **do**
4:     Set $\mathcal{B}_{i,p} = \{l_{i,2^{p-2}}, \ldots, l_{i,2^{p-1}-1}\}$
5:     Set $k = 1$ and $x_{i,p,1} = x_{i,p-1,K}$
6:     $\{v_{i,j,s}\}_{j\in[T_1],s\in\{0,1\}^{\leq j}} = \text{DP-FW}\left(\mathcal{B}_{i,p}, T_1, b\right)$
7:     **for all** leaf vertices $s$ reached in DP-FW **do**
8:         **Communicate to server:** $\{\langle c_n, v_{i,j,s}\rangle + \xi_{i,n}\}_{n\in[d]}$, where $\xi_{i,n} \sim \text{Lap}(\lambda_{i,j,s})$
9:         **Receive from server:** $\bar{w}_{j,s}$
10:        Update $x_{i,p,k}$ according to Equation (2)
11:        Update $k = k + 1$
12:     **end for**
13:     Final iterate outputs $x_{i,p,K}$
14:     **for** $t = 2^{p-1}$ to $2^p - 1$ **do**
15:        Receive loss $l_{i,t} : \mathcal{X} \to \mathbb{R}$ and pay cost $l_{i,t}(x_{i,p,K})$
16:     **end for**
17: **end for**

---

## 3.3 Theoretical Guarantees

Now we are ready to present theoretical guarantees for Fed-DP-OPE-Stoch.

**Theorem 1.** *Assume that loss function $l_{i,t}(\cdot)$ is convex, $\alpha$-Lipschitz, $\beta$-smooth w.r.t. $\|\cdot\|_1$. Setting $\lambda_{i,j,s} = \frac{4\alpha 2^j}{b\varepsilon}$, $b = \frac{2^{p-1}}{(p-1)^2}$ and $T_1 = \frac{1}{2}\log\left(\frac{b\varepsilon\beta\sqrt{m}}{\alpha\log d}\right)$, Fed-DP-OPE-Stoch (i) satisfies $\varepsilon$-DP and (ii) achieves the per-client regret of*

$$O\left((\alpha + \beta)\log T\sqrt{\frac{T\log d}{m}} + \frac{\sqrt{\alpha\beta}\log d\sqrt{T}\log T}{m^{\frac{1}{4}}\sqrt{\varepsilon}}\right)$$

---

**Algorithm 2** Fed-DP-OPE-Stoch: Central server

---

1: **Input:** Phases $P$, number of clients $m$, decision set $\mathcal{X} = \Delta_d$ with vertices $\{c_1, \ldots, c_d\}$.
2: **Initialize:** Pick any $z \in \mathcal{X}$ and broadcast to clients.
3: **for** $p = 2$ to $P$ **do**
4:    **Receive from clients:** $\{\langle c_n, v_{i,j,s} \rangle + \xi_{i,n}\}_{n \in [d]}$
5:    Update $\bar{w}_{j,s}$ according to Equation (1)
6:    **Communicate to clients:** $\bar{w}_{j,s}$
7: **end for**

---

with (iii) a communication cost of $O\left(m^{\frac{5}{4}} d \sqrt{\frac{T \varepsilon \beta}{\alpha \log d}}\right)$.

A similar result characterizing the performance of Fed-DP-OPE-Stoch under $(\varepsilon, \delta)$-DP constraint is shown in Theorem 6.

*Proof sketch of DP guarantees and communication costs.* Given neighboring datasets $\mathcal{S}$ and $\mathcal{S}'$ differing in $l_{i_1, t_1}$ or $l'_{i_1, t_1}$, we first note that $l_{i_1, t_1}$ or $l'_{i_1, t_1}$ is used in only one phase, denoted as $p_1$. Furthermore, note that $l_{i_1, t_1}$ or $l'_{i_1, t_1}$ is used in the gradient computation for at most $2^{j-|s|}$ iterations, corresponding to descendant leaf nodes. This insight allows us to set noise levels $\lambda_{i,j,s}$ sufficiently large to ensure each of these iterations is $\varepsilon/2^{j-|s|}$-DP regarding output $\{\langle c_n, v_{i_1,j,s} \rangle\}_{n \in [d], j \in [T_1], |s|=j}$. By basic composition and post-processing, we can make sure the final output is $\varepsilon$-DP.

The communication cost is obtained by observing that there are $p$ phases, and within each phase, there are $O(2^{T_1})$ leaf vertices. Thus, the communication frequency scales in $O(\sum_p 2^{T_1})$ and the communication cost scales in $O(md \sum_p 2^{T_1})$. $\qquad\square$

*Proof sketch of regret upper bound.* We first give bounds for the total regret in phase $p \in [P]$. We can show that for every $t$ in phase $p$, we have

$$L_t(x_{i,p,k+1}) \leq L_t(x_{i,p,k}) + \eta_{i,p,k}[L_t(x^\star) - L_t(x_{i,p,k})] + \eta_{i,p,k} \|\nabla L_t(x_{i,p,k}) - \bar{v}_{p,k}\|_\infty$$
$$+ \eta_{i,p,k}\left(\langle \bar{v}_{p,k}, \bar{w}_{p,k} \rangle - \min_{w \in \mathcal{X}}\langle \bar{v}_{p,k}, w \rangle\right) + \frac{1}{2}\beta \eta_{i,p,k}{}^2.$$

To upper bound the regret, we bound the following two quantities separately.

**Step 1: bound** $\|\nabla L_t(x_{i,p,k}) - \bar{v}_{p,k}\|_\infty$ **in Lemma 3.** We show that every index of the $d$-dimensional vector $\nabla L_t(x_{i,p,k}) - \bar{v}_{p,k}$ is $O(\frac{\alpha^2+\beta^2}{bm})$-sub-Gaussian by induction on the depth of vertex in Lemma 2. Therefore, $\mathbb{E}[\|\nabla L_t(x_{i,p,k}) - \bar{v}_{p,k}\|_\infty]$ is upper bounded by $O\left((\alpha+\beta)\sqrt{\frac{\log d}{bm}}\right)$.

**Step 2: bound** $\langle \bar{v}_{p,k}, \bar{w}_{p,k} \rangle - \min_{w \in \mathcal{X}}\langle \bar{v}_{p,k}, w \rangle$ **in Lemma 5.** We establish that $\langle \bar{v}_{p,k}, \bar{w}_{p,k} \rangle - \min_{w \in \mathcal{X}}\langle \bar{v}_{p,k}, w \rangle$ is upper bounded by $\frac{2}{m}\max_{n:1 \leq n \leq d}\left|\sum_{i=1}^m \xi_{i,n}\right|$. This bound incorporates the maximum absolute sum of $m$ IID Laplace random variables $\xi_{i,n}$. To quantify the bound on the sum of $m$ IID Laplace random variables, we refer to Lemma 4. Consequently, we have $\mathbb{E}\left[\langle \bar{v}_{p,k}, \bar{w}_{p,k} \rangle - \min_{w \in \mathcal{X}}\langle \bar{v}_{p,k}, w \rangle\right]$ upper bounded by $O\left(\frac{\lambda_{i,j,s}\ln d}{\sqrt{m}}\right)$.

With the two steps above, we can show that for each time step $t$ in phase $p$, $\mathbb{E}[L_t(x_{i,p,K}) - L_t(x^\star)]$ is upper bounded by $O\left((\alpha+\beta)\sqrt{\frac{\log d}{2^p m}} + \frac{\sqrt{\alpha\beta}\log d}{m^{\frac{1}{4}}\sqrt{2^p \varepsilon}}\right)$. Summing up all the phases gives us the final upper bound. The full proof of Theorem 1 can be found in Appendix C.2. $\qquad\square$

When $\beta$ is small, Theorem 1 reduces to the following corollary.

**Corollary 1.** *If $\beta = O(\frac{\log d}{\sqrt{m}T\varepsilon})$, then, Fed-DP-OPE-Stoch (i) satisfies $\varepsilon$-DP and (ii) achieves the per-client regret of*

$$Reg(T,m) = O\left( \sqrt{\frac{T \log d}{m}} + \frac{\log T \log d}{\sqrt{m}\varepsilon} \right),$$

*with (iii) a communication cost of $O\left(md \log T\right)$.*

The full proof of Corollary 1 can be found in Appendix C.2.

*Remark* 2. We note that the regret for the single-player counterpart Asi et al. (2022b) scales in $O\left( \sqrt{T \log d} + \frac{\log d \log T}{\varepsilon} \right)$ when $\beta = O(\frac{\log d}{T\varepsilon})$. Compared with this upper bound, Fed-DP-OPE-Stoch achieves $\sqrt{m}$-fold speed-up, with a communication cost of $O(md \log T)$. This observation underscores the learning performance acceleration due to collaboration. Furthermore, our approach extends beyond the mere privatization of selected experts; we ensure the privacy of all information exchanged between the central server and clients.

*Remark* 3. We remark that Fed-DP-OPE-Stoch can be slightly modified into a centrally differentially private algorithm, assuming client-server communication is secure. Specifically, we change the local privatization process to a global privatization process on the server side. This mechanism results in less noise added and thus better utility performance. The detailed design is deferred to Appendix C.4 and the theoretical guarantees are provided in Theorem 7.

## 4 Federated OPE with Oblivious Adversaries: Lower bounds

In this section, we shift our attention to the more challenging oblivious adversaries. We establish new lower bounds for general oblivious adversaries.

To provide a foundational understanding, we start with some intuition behind FL. FL can potentially speed up the learning process by collecting more data at the same time to gain better insights to future predictions. In the stochastic setting, the advantage of collaboration lies in the ability to collect more observations from the same distribution, which leads to variance reduction. However, when facing oblivious adversaries, the problem changes fundamentally. Oblivious adversaries can select loss functions arbitrarily, meaning that having more data does not necessarily help with predicting their future selections.

The following theorem formally establishes the aforementioned intuition.

**Theorem 2.** *For any federated OPE algorithm against oblivious adversaries, the per-client regret is lower bounded by $\Omega(\sqrt{T \log d})$. Let $\varepsilon \in (0,1]$ and $\delta = o(1/T)$, for any $(\varepsilon, \delta)$-DP federated OPE algorithm, the per-client regret is lower bounded by $\Omega\left( \min\left( \frac{\log d}{\varepsilon}, T \right) \right).$*

*Remark* 4. Theorem 2 states that with oblivious adversaries, the per-client regret under any federated OPE is fundamentally independent of the number of clients $m$, indicating that, the collaborative effort among multiple clients does not yield a reduction in the regret lower bound. This is contrary to typical scenarios where collaboration can lead to shared insights and improve overall performance. The varied and unpredictable nature of oblivious adversaries nullifies the typical advantages of collaboration. Theorem 2 also emphasizes the influence of the DP guarantees. Our lower bounds represent the first non-trivial impossibility results for the federated OPE to the best of our knowledge.

*Proof sketch.* We examine the case when *all clients receive the same loss function from the oblivious adversary* at each time step, i.e. $l_{i,t} = l'_t$. Within this framework, we define the *"average policy"* among all clients, i.e., $p'_t(k) = \frac{1}{m}\sum_{i=1}^m p_{i,t}(k), \forall k \in [d]$. This leads us to express the per-client regret as: $\text{Reg}(T,m) = \sum_{t=1}^T \sum_{k=1}^d p'_t(k) \cdot l'_t(k) - \sum_{t=1}^T l'_t(x^\star)$. Note that $p'_t(k)$ is defined by $p_{1,t}(k), \ldots, p_{m,t}(k)$, which in turn are determined by $l_{1,1}, \ldots, l_{m,t-1}$. According to our choice of $l_{i,t} = l'_t$, $p'_t(k)$ is determined by $l'_1, l'_2, \ldots, l'_{t-1}$. Therefore, $p'_1, p'_2, \ldots, p'_t$ are generated by a legitimate algorithm for online learning with expert advice problems. Through our **policy reduction approach in FL**, we can reduce the federated problem to a single-player setting, showing the equivalence of per-client and single-player regret against the oblivious adversary we construct, thus obtaining the regret lower bound. We believe that this technique will be useful in future analysis of other FL algorithm. Incorporating DP constraints, we refer to Lemma 9, which transforms the DP

online learning problem to a well-examined DP batch model. The full proof of Theorem 2 can be found in Appendix D. □

# 5 Federated OPE with Oblivious Adversaries: Realizable Setting

Given the impossibility results in the general oblivious adversaries setting, one natural question we aim to answer is, *is there any special scenarios where we can still harness the power of federation even in presence of oblivious adversaries?* Towards this end, in this section, we focus on the (near) realizable setting, formally defined below.

**Definition 2** (Realizability). A federated OPE problem is *realizable* if there exists a feasible solution $x^\star \in [d]$ such that $\sum_{t=1}^{T} l_{i,t}(x^\star) = 0, \forall i \in [m]$. If the best expert achieves small loss $L^\star \ll T$, i.e., there exists $x^\star \in [d]$ such that $\sum_{t=1}^{T} l_{i,t}(x^\star) \leq L^\star, \forall i \in [m]$, the problem is near-realizable.

Intuitively, collaboration is notably advantageous in this context, as all clients share the same goal of reaching the zero-loss solution $x^\star$. As more clients participate, the shared knowledge pool expands, making the identification of the optimal solution more efficient. In the following, we first provide regret lower bounds to quantify this benefit formally, and then show that a sparse vector based federated algorithm can almost achieve such lower bounds thus is nearly optimal.

## 5.1 Lower Bound

**Theorem 3.** *Let $\varepsilon \leq 1$ and $\delta \leq \varepsilon/d$. For any $(\varepsilon, \delta)$-DP federated OPE algorithm against oblivious adversaries in the realizable setting, the per-client regret is lower bounded by $\Omega\left(\frac{\log(d)}{m\varepsilon}\right)$.*

*Remark* 5. In the single-player setting, the regret bound is $\Omega\left(\frac{\log(d)}{\varepsilon}\right)$ (Asi et al., 2023). In the federated setting, our results imply that the per-client regret is reduced to $\frac{1}{m}$ times the single-player regret. This indicates a possible $m$-fold speed-up in the federated setting.

*Proof sketch.* To begin, we consider a specific oblivious adversary. We introduce two prototype loss functions: $l^0(x) = 0$ for all $x \in [d]$ and for $j \in [d]$ $l^j(x) = 0$ if $x = j$ and otherwise $l^j(x) = 1$. An oblivious adversary picks one of the $d$ sequences $\mathcal{S}^1, \ldots, \mathcal{S}^d$ uniformly at random, where $\mathcal{S}^j = (l_{1,1}^j, \ldots, l_{m,T}^j)$ such that $l_{i,t}^j = l^0$ if $t = 1, \ldots, T - k$, otherwise, $l_{i,t}^j = l^j$, and $k = \frac{\log d}{2m\varepsilon}$. Assume there exists an algorithm such that the per-client regret is upper bounded by $\frac{\log(d)}{32m\varepsilon}$. This would imply that for at least $d/2$ of $\mathcal{S}^1, \ldots, \mathcal{S}^d$, the per-client regret is upper bounded by $\frac{\log(d)}{16m\varepsilon}$. Assume without loss of generality these sequences are $\mathcal{S}^1, \ldots, \mathcal{S}^{d/2}$. We let $\mathcal{B}_j$ be the set of expert selections that has low regret on $\mathcal{S}^j$. We can show that $\mathcal{B}_j$ and $\mathcal{B}_{j'}$ are disjoint for $j \neq j'$, implying *choosing any expert $j$ leads to low regret for loss sequence $\mathcal{S}^j$ but high regret for $\mathcal{S}^{j'}$.* By group privacy, we have $\mathbb{P}\left(\mathcal{A}\left(\mathcal{S}^j\right) \in \mathcal{B}_{j'}\right) \geq e^{-mk\varepsilon} \mathbb{P}\left(\mathcal{A}\left(\mathcal{S}^{j'}\right) \in \mathcal{B}_{j'}\right) - mk\delta$, leading to a contradiction when $d \geq 32$. The full proof of Theorem 3 can be found in Appendix E.1. □

## 5.2 Algorithm Design

We develop a new algorithm Fed-SVT. Our algorithm operates as follows:

**Periodic communication.** We adopt a fixed communication schedule in our federated setting, splitting the time horizon $T$ into $T/N$ phases, each with length $N$. In Fed-SVT, every client selects the same expert, i.e., $x_{i,t} = x_t$ at each time step $t$. Initially, each client starts with a randomly chosen expert $x_1$. At the beginning of each phase $n$, each client sends the accumulated loss of the last phase $\sum_{t'=(n-1)N}^{nN-1} l_{i,t'}(x)$ to the central server.

**Global expert selection.** The server, upon receiving $\sum_{t'=(n-1)N}^{nN-1} l_{i,t'}(x)$ from all clients, decides whether to continue with the current expert or switch to a new one. This decision is grounded in the Sparse-Vector algorithm (Asi et al., 2023), where the accumulated loss from all clients over a phase is treated as a single loss instance in the Sparse-Vector algorithm. Based on the server's expert decision, clients update their experts accordingly. The full algorithm is provided in Appendix E.2.

### 5.3 Theoretical Guarantees

**Theorem 4.** *Let $l_{i,t} \in [0,1]^d$ be chosen by an oblivious adversary under near-realizability assumption. Fed-SVT is $\varepsilon$-DP, the communication cost scales in $O\left(mdT/N\right)$, and with probability at least $1 - O(\rho)$, the pre-client regret is upper bounded by*
$$O\left( \frac{\log^2(d) + \log\left(\frac{T^2}{N^2\rho}\right)\log\left(\frac{d}{\rho}\right)}{m\varepsilon} + (N + L^\star)\log\left(\frac{d}{\rho}\right)\right).$$
*Moreover, Fed-SVT is $(\varepsilon, \delta)$-DP, the communication cost scales in $O\left(mdT/N\right)$, and with probability at least $1 - O(\rho)$, the pre-client regret is upper bounded by* $O\left( \frac{\log^{\frac{3}{2}}(d)\sqrt{\log(\frac{1}{\delta})} + \log\left(\frac{T^2}{N^2\rho}\right)\log\left(\frac{d}{\rho}\right)}{m\varepsilon} + (N + L^\star)\log\left(\frac{d}{\rho}\right)\right).$

*Remark* 6. Note that when $N + L^\star = O\left(\frac{\log T}{m\varepsilon}\right)$, then, under Fed-SVT, the per-client regret upper bound scales in $O\left(\frac{\log^2 d + \log T \log d}{m\varepsilon}\right)$ for $\varepsilon$-DP and $O\left(\frac{\log T \log d + \log^{3/2} d\sqrt{\log(1/\delta)}}{m\varepsilon}\right)$ for $(\varepsilon, \delta)$-DP. Compared to the best upper bound $O\left(\frac{\log^2 d + \log T \log d}{\varepsilon}\right)$ and $O\left(\frac{\log T \log d + \log^{3/2} d\sqrt{\log(1/\delta)}}{\varepsilon}\right)$ for the single-player scenario (Asi et al., 2023), our results indicate an $m$-fold regret speed-up. Note that our lower bound (Theorem 3) scales in $\Omega\left(\frac{\log d}{m\varepsilon}\right)$. Our upper bound matches with the lower bound in terms of $m$ and $\epsilon$, and is nearly optimal up to logarithmic factors.

The proof of Theorem 4 is presented in Appendix E.3.

## 6 Numerical Experiments

**Fed-DP-OPE-Stoch.** We conduct experiments in a synthetic environment to validate the theoretical performances of Fed-DP-OPE-Stoch, and compare them with its single-player counterpart Limited Updates (Asi et al., 2022b).

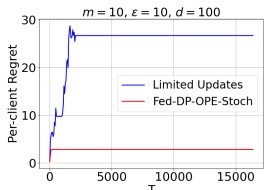

Figure 1: Per-client regret.

We first generate true class distributions for each client $i \in [m]$ and timestep $t \in [T]$, which are sampled IID from Gaussian distributions with means and variances sampled uniformly. Following Gaussian sampling, we apply a softmax transformation and normalization to ensure the outputs are valid probability distributions. Then we generate a sequence of IID loss functions $l_{1,1}, \ldots, l_{m,T}$ for $m$ clients over $T$ timesteps using cross-entropy between true class distributions and predictions. We set $T_1 = 1$ in Fed-DP-OPE-Stoch, therefore the communication cost scales in $O(md \log T)$. We set $m = 10$, $T = 2^{14}$, $\varepsilon = 10$, $\delta = 0$ and $d = 100$. The per-client cumulative regret as a function of $T$ is plotted in Figure 1. We see that Fed-DP-OPE-Stoch outperforms Limited Updates significantly, indicating the regret speed-up due to collaboration. Results with different seeds are provided in Appendix F.

**Fed-SVT.** We conduct experiments in a synthetic environment, comparing Fed-SVT with the single-player model Sparse-Vector (Asi et al., 2023).

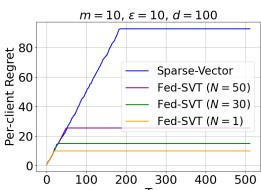

Figure 2: Per-client regret.

We generate random losses for each expert at every timestep and for each client, ensuring that one expert always has zero loss to simulate an optimal choice. We set $m = 10$, $T = 2^9$, $\varepsilon = 10$, $\delta = 0$ and $d = 100$. In Fed-SVT, we experiment with communication intervals $N = 1, 30, 50$, where communication cost scales in $O(mdT/N)$. The per-client cumulative regret as a function of $T$ is plotted in Figure 2. Our results show that Fed-SVT significantly outperforms the Sparse-Vector, highlighting the benefits of collaborative expert selection in regret speed-up, even at lower communication costs (notably in the $N = 50$ case). Results with different seeds are provided in Appendix F. Additionally, we evaluate the performances of Fed-SVT on the MovieLens-1M dataset (Harper and Konstan, 2015) in Appendix G.

## 7 Conclusions

In this paper, we have advanced the state-of-the-art of differentially private federated online prediction from experts, addressing both stochastic and oblivious adversaries. Our Fed-DP-OPE-Stoch algorithm

showcases a significant $\sqrt{m}$-fold regret speed-up compared to single-player models with stochastic adversaries, while effectively maintaining logarithmic communication costs. For oblivious adversaries, we established non-trivial lower bounds, highlighting the limited benefits of client collaboration. Additionally, our Fed-SVT algorithm demonstrates an $m$-fold speed-up, indicating near-optimal performance in settings with low-loss experts. One limitation of this work is the lack of experiments on real-world federated learning scenarios such as recommender systems and healthcare. We leave this exploration to future work.

## Acknowledgments

The work of F. Gao, R. Huang and J. Yang was supported in part by the U.S. National Science Foundation under the grants CNS-1956276, CNS-2114542 and ECCS-2133170.

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

# A Related Work

**Online learning with stochastic adversaries.** Online learning with stochastic adversaries has been extensively studied (Rakhlin et al., 2011; Duchi et al., 2011; Hu et al., 2009; Li et al., 2020a; Caron et al., 2012; Buccapatnam et al., 2014; Tossou et al., 2017; Wu et al., 2015). This problem is also closely related to stochastic convex optimization (Shalev-Shwartz et al., 2009; Mahdavi et al., 2013; Duchi et al., 2015; Agarwal et al., 2011; Duchi et al., 2016) since online learning with stochastic adversaries can be transformed into the stochastic convex optimization problem using online-to-batch conversion. In the federated setting, Mitra et al. (2021) proposed a federated online mirror descent method. Patel et al. (2023) presented a federated projected online stochastic gradient descent algorithm. Research on online learning with stochastic adversaries incorporating DP has also seen significant advancements (Bassily et al., 2014, 2019; Feldman et al., 2020; Asi et al., 2021a,b, 2022b).

**Online learning with non-stochastic adversaries.** Online learning with non-stochastic adversaries has been extensively studied (Cesa-Bianchi and Lugosi, 2006; Littlestone and Warmuth, 1994; Freund and Schapire, 1997; Zinkevich, 2003). Federated online learning with non-stochastic adversaries, a more recent development, was introduced by Hong and Chae (2021). Mitra et al. (2021) developed a federated online mirror descent method. Park et al. (2022) presented a federated online gradient descent algorithm. Kwon et al. (2023) studied data heterogeneity. Furthermore, Li et al. (2018) and Patel et al. (2023) explored distributed online and bandit convex optimization. Gauthier et al. (2023) focused on the asynchronous settings. Research into online learning with limited switching has also been explored by Kalai and Vempala (2005); Geulen et al. (2010); Altschuler and Talwar (2018); Chen et al. (2020); Sherman and Koren (2021); He et al. (2022); Li et al. (2022b); Cheng et al. (2023).

Differentially private online learning with non-stochastic adversaries was pioneered by Dwork et al. (2010). Several studies have explored OPE problem with DP constraints (Jain et al., 2012; Guha Thakurta and Smith, 2013; Jain and Thakurta, 2014; Agarwal and Singh, 2017; Asi et al., 2022b). Specifically, Jain and Thakurta (2014) and Agarwal and Singh (2017) focused on methods based on follow the regularized leader. Asi et al. (2022b) proposed an algorithm using the shrinking dartboard method.

Studies addressing differentially private online learning with non-stochastic adversaries in other contexts have also made significant strides. Kaplan et al. (2023b) focused on online classification problem with joint DP constraints. Fichtenberger et al. (2023) introduced a constant improvement. Gonen et al. (2019) and Kaplan et al. (2023a) studied the relationship between private learning and online learning. Agarwal et al. (2023a) and Agarwal et al. (2023b) studied the DP online convex optimization problem. Research on differentially private federated online learning with non-stochastic adversaries is very limited to the best of our knowledge.

**Online learning under realizability assumption.** Several works have studied online learning in the realizable setting. Srebro et al. (2010) studied mirror descent for online optimization. Shalev-Shwartz et al. (2012) studied the weighted majority algorithm. There is also a line of work studying stochastic convex optimization in the realizable setting (Cotter et al., 2011; Ma et al., 2018; Vaswani et al., 2019; Liu and Belkin, 2018; Woodworth and Srebro, 2021; Asi et al., 2022a). Additionally, some researchers have recently focused on the variance or the path length of the best expert (Cesa-Bianchi et al., 2007; Steinhardt and Liang, 2014; Wei and Luo, 2018; Bubeck et al., 2019). When DP constraint is considered, Asi et al. (2023) introduced algorithms using the sparse vector technique. Golowich and Livni (2021) studied DP online classification.

**Multi-armed bandits with DP.** Research on DP multi-armed bandits has developed along various lines (Mishra and Thakurta, 2015; Tossou and Dimitrakakis, 2016; Sajed and Sheffet, 2019; Azize and Basu, 2022). Further, Hu and Hegde (2022) proposed a Thompson-sampling-based approach, Tao et al. (2022) explored the heavy-tailed rewards scenario, and Chowdhury and Zhou (2022a) achieved optimal regret in a distributed setting. Additionally, Ren et al. (2020) and Zheng et al. (2020) investigated local DP constraints. For linear contextual bandits with DP, significant studies were conducted by Shariff and Sheffet (2018); Wang et al. (2020, 2022b) and Hanna et al. (2022). Furthermore, Hanna et al. (2022); Chowdhury and Zhou (2022b); Garcelon et al. (2022) and Tenenbaum et al. (2021) focused on the shuffle model. Li et al. (2023) studied kernelized bandits with distributed biased feedback. Wu et al. (2023) and Charisopoulos et al. (2023) tackled privacy and robustness simultaneously.

**Federated multi-armed bandits.** Federated bandits have been studied extensively recently. Shi and Shen (2021) and Shi et al. (2021) investigated federated stochastic multi-armed bandits without and with personalization, respectively. Wang et al. (2019) considered the distributed setting. Huang et al. (2021), Wang et al. (2022a) and Li and Wang (2022) studied federated linear contextual bandits. Li et al. (2022b) focused on federated $\mathcal{X}$-armed bandits problem. Additionally, Cesa-Bianchi et al. (2016), Bar-On and Mansour (2019), Yi and Vojnovic (2022) and Yi and Vojnovic (2023) have studied cooperative multi-armed bandits problem with data exchange among neighbors. M Ghari and Shen (2022) have explored online model selection where each client learns a kernel-based model, utilizing the specific characteristics of kernel functions.

When data privacy is explicitly considered, Li et al. (2020b) and Zhu et al. (2021) studied federated bandits with DP guarantee. Dubey and Pentland (2022) investigated private and byzantine-proof cooperative decision-making in the bandits setting. Dubey and Pentland (2020) and Zhou and Chowdhury (2023) considered the linear contextual bandit model with joint DP guarantee. Li et al. (2022a) studied private distributed linear bandits with partial feedback. Huang et al. (2023) studied federated linear contextual bandits with user-level DP guarantee.

# B Applications of Differentially Private Federated OPE

Differentially private federated online prediction has many important real-world applications. We provide three examples below.

**Personalized Healthcare:** Consider a federated online prediction setting where patients' wearable devices collect and process health data locally, and the central server aggregates privacy-preserving updates from devices to provide health recommendations or alerts. DP federated online prediction can speed up the learning process and improve prediction accuracy without exposing individual users' health data, thus ensuring patient privacy.

**Financial Fraud Detection:** DP federated online prediction can also enhance fraud detection systems across banking and financial services. Each client device (e.g. PC) locally analyzes transaction patterns and flags potential fraud without revealing sensitive transaction details to the central server. The server's role is to collect privacy-preserving updates from these clients to improve the global fraud detection model. This method ensures that the financial company can dynamically adapt to new fraudulent tactics, improving detection rates while safeguarding customers' financial privacy.

**Personalized Recommender Systems:** Each client (e.g. smartphone) can personalize content recommendations by analyzing user interactions and preferences locally. The central server (e.g. company) aggregates privacy-preserving updates from all clients to refine the recommendation model. Thus, DP federated online prediction improves the whole recommender system performance while maintaining each client's privacy.

# C Algorithms and Proofs for Fed-DP-OPE-Stoch

## C.1 DP-FW

In Fed-DP-OPE-Stoch, we run a DP-FW subroutine (Asi et al., 2021b) at each client $i$ in each phase $p$. DP-FW maintains $T_1$ binary trees indexed by $1 \leq j \leq T_1$, each with depth $j$, where $T_1$ is a predetermined parameter. An example of the tree structure is shown below. We introduce the notation $s \in \{0,1\}^{\leq j}$ to denote vertices within binary tree $j$. $\emptyset$ signifies the tree's root. For any $s, s' \in \{0,1\}^{\leq j}$, if $s = s'0$, then $s$ denotes the left child of $s'$. Conversely, when $s = s'1$, $s$ is attributed as the right child of $s'$. For each client $i$, each vertex $s$ in the binary tree $j$ corresponds to a parameter $x_{i,j,s}$ and a gradient estimate $v_{i,j,s}$. Each client iteratively updates parameters and gradient estimates by visiting the vertices of a binary tree according to the Depth-First Search (DFS) order: as it visits a left child vertex $s$, the algorithm maintains the parameter $x_{i,j,s}$ and the gradient estimate $v_{i,j,s}$ identical to those of its parent vertex $s'$, i.e. $v_{i,j,s} = v_{i,j,s'}$ and $x_{i,j,s} = x_{i,j,s'}$. As it proceeds to a right child vertex, we uniformly select $2^{-|s|}b$ loss functions from set $\mathcal{B}_{i,p} = \{l_{i,2^{p-2}}, \ldots, l_{i,2^{p-1}-1}\}$ to subset $\mathcal{B}_{i,j,s}$ without replacement, where $b$ denoting the predetermined batch size and $|s|$ representing the depth of the vertex $s$. Then the algorithm improves the gradient estimate $v_{i,j,s}$ at the current vertex $s$ of tree $j$ using the estimate $v_{i,j,s'}$ at the parent vertex $s'$, i.e.

$$v_{i,j,s} = v_{i,j,s'} + \nabla l(x_{i,j,s}; \mathcal{B}_{i,j,s}) - \nabla l(x_{i,j,s'}; \mathcal{B}_{i,j,s}). \tag{3}$$

---

**Algorithm 3** DP-FW at client $i$ (Asi et al., 2021b)

---

1: **Input:** Sample set $\mathcal{B}$, number of trees $T_1$, batch size $b$.
2: **for** $j = 1$ to $T_1$ **do**
3:     Set $x_{i,j,\emptyset} = x_{i,j-1,L_{j-1}}$
4:     Uniformly select $b$ samples to $\mathcal{B}_{i,j,\emptyset}$
5:     $v_{i,j,\emptyset} = \nabla l_{i,t}(x_{i,j,\emptyset}; \mathcal{B}_{i,j,\emptyset})$
6:     **for** $s \in \mathrm{DFS}[j]$ **do**
7:         Let $s = s'a$ where $a \in \{0, 1\}$
8:         **if** $a == 0$ **then**
9:             $v_{i,j,s} = v_{i,j,s'}; x_{i,j,s} = x_{i,j,s'}$
10:        **else**
11:           Uniformly select $2^{-|s|}b$ samples to $\mathcal{B}_{i,j,s}$
12:           Update $v_{i,j,s}$ according to Equation (3)
13:        **end if**
14:     **end for**
15: **end for**
16: **Return** $\{v_{i,j,s}\}_{j \in [T_1], s \in \{0,1\}^{\leq j}}$

---

where $\nabla l(\cdot; \mathcal{B}_{i,j,s}) = \frac{1}{|\mathcal{B}_{i,j,s}|} \sum_{l \in \mathcal{B}_{i,j,s}} \nabla l(\cdot)$, and $\mathcal{B}_{i,j,s}$ is the subset of loss functions at vertex $s$ in the binary tree $j$ for client $i$. The full algorithm is shown in Algorithm 3.

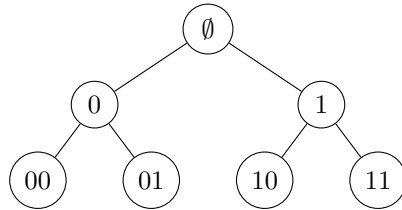

Figure 3: Binary tree with depth $j = 2$ in Algorithm 3.

## C.2   Proof of Theorem 1 and Corollary 1 (for pure DP)

**Theorem 5** (Restatement of Theorem 1). *Assume that loss function $l_{i,t}(\cdot)$ is convex, $\alpha$-Lipschitz, $\beta$-smooth w.r.t. $\|\cdot\|_1$. Setting $\lambda_{i,j,s} = \frac{4\alpha 2^j}{b\varepsilon}$, $b = \frac{2^{p-1}}{(p-1)^2}$ and $T_1 = \frac{1}{2} \log\left(\frac{b\varepsilon\beta\sqrt{m}}{\alpha \log d}\right)$, Fed-DP-OPE-Stoch (i) satisfies $\varepsilon$-DP and (ii) achieves the per-client regret of*

$$O\left((\alpha + \beta) \log T \sqrt{\frac{T \log d}{m}} + \frac{\sqrt{\alpha\beta} \log d \sqrt{T} \log T}{m^{\frac{1}{4}} \sqrt{\varepsilon}}\right)$$

*with (iii) a communication cost of $O\left(m^{\frac{5}{4}} d \sqrt{\frac{T\varepsilon\beta}{\alpha \log d}}\right)$.*

We define population loss as $L_t(x) = \mathbb{E}[\frac{1}{m} \sum_{i=1}^m l_{i,t}(x_{i,t})]$. The per-client regret can be expressed as $\mathbb{E}\left[\sum_{t=1}^T L_t(x_{i,t}) - \min_{x^\star \in \mathcal{X}} \sum_{t=1}^T L_t(x^\star)\right]$. We use $v_{i,p,k}, w_{i,p,k}, x_{i,p,k}$ and $\eta_{i,p,k}$ to denote quantities corresponding to phase $p$, iteration $k$, and client $i$. Also we introduce some average quantities $\bar{v}_{p,k} = \frac{1}{m} \sum_{i=1}^m v_{i,p,k}$ and $\bar{w}_{p,k} = \frac{1}{m} \sum_{i=1}^m w_{i,p,k}$.

To prove the theorem, we start with Lemma 1 that gives pure privacy guarantees.

**Lemma 1.** *Assume that $2^{T_1} \leq b$. Setting $\lambda_{i,j,s} = \frac{4\alpha 2^j}{b\varepsilon}$, Fed-DP-OPE-Stoch is $\varepsilon$-DP.*

*Proof.* Let $\mathcal{S}$ and $\mathcal{S}'$ be two neighboring datasets and assume that they differ in sample $l_{i_1,t_1}$ or $l'_{i_1,t_1}$, where $2^{p_1-1} \leq t_1 < 2^{p_1}$. Let $\mathcal{B}_{i_1,j,s}$ be the set that contains $l_{i_1,t_1}$ or $l'_{i_1,t_1}$. Recall that $|\mathcal{B}_{i_1,j,s}| = 2^{-|s|}b$. The key point is that this set is used in the calculation of $v_{i_1,p_1,k}$ for at most $2^{j-|s|}$

iterates, i.e. the leaves that are descendants of the vertex. Let $k_0$ and $k_1$ be the first and last iterate such that $\mathcal{B}_{i_1,j,s}$ is used for the calculation of $v_{i_1,p_1,k}$, hence $k_1 - k_0 + 1 \leq 2^{j-|s|}$.

First, we show that $(\langle c_n, v_{1,1,1}\rangle, \ldots, \langle c_n, v_{m,P,K}\rangle) \approx_{(\varepsilon,0)} (\langle c_n, v'_{1,1,1}\rangle, \ldots, \langle c_n, v'_{m,P,K}\rangle)$ holds for $1 \leq n \leq d$. For our purpose, it suffices to establish that $(\langle c_n, v_{i_1,p_1,k_0}\rangle, \ldots, \langle c_n, v_{i_1,p_1,k_1}\rangle) \approx_{(\varepsilon,0)}$ $(\langle c_n, v'_{i_1,p_1,k_0}\rangle, \ldots, \langle c_n, v'_{i_1,p_1,k_1}\rangle)$ holds for $1 \leq n \leq d$, since $l_{i_1,t_1}$ or $l'_{i_1,t_1}$ is only used in client $i_1$, at phase $p_1$ and iteration $k_0, \ldots, k_1$. Therefore it is enough to show that $\langle c_n, v_{i_1,p_1,k}\rangle \approx_{(\frac{\varepsilon}{2^{j-|s|}},0)}$ $\langle c_n, v'_{i_1,p_1,k}\rangle$ holds for $k_0 \leq k \leq k_1$ and $1 \leq n \leq d$, because $k_1 - k_0 + 1 \leq 2^{j-|s|}$. Note that $|\langle c_n, v_{i_1,p_1,k} - v'_{i_1,p_1,k}\rangle| \leq \frac{4\alpha}{2^{-|s|}b}$. Setting $\lambda_{i,j,s} = \frac{4\alpha 2^j}{b\varepsilon}$ and applying standard results of Laplace mechanism (Dwork et al. (2014)) lead to our intended results. Finally, we can show that $(x_{1,1,K}, \ldots, x_{m,P,K}) \approx_{(\varepsilon,0)} (x'_{1,1,K}, \ldots, x'_{m,P,K})$ by post-processing. $\qquad\square$

To help prove the upper bound of the regret, we introduce some lemmas first.

**Lemma 2.** *Let $t$ be the time-step, $p$ be the index of phases, $i$ be the index of the clients, and $(j,s)$ be a vertex. For every index $1 \leq k \leq d$ of the vectors, we have*

$$\mathbb{E}\left[\exp\left\{c(\bar{v}_{p,j,s,k} - \nabla L_{t,k}(x_{i,p,j,s}))\right\}\right] \leq \exp\left(\frac{O(1)c^2(\alpha^2 + \beta^2)}{bm}\right),$$

*where $\bar{v}_{p,j,s} = \frac{1}{m}\sum_{i=1}^{m} v_{i,p,j,s}$.*

*Proof.* Let us fix $p$, $t$, $k$ and $i$ for simplicity and let $A_{j,s} = \bar{v}_{p,j,s,k} - \nabla L_{t,k}(x_{i,p,j,s})$. We prove the lemma by induction on the depth of the vertex, i.e., $|s|$. If $|s| = 0$, then $v_{i,p,j,\emptyset} = \nabla l(x_{i,p,j,\emptyset}; \mathcal{B}_{i,p,j,\emptyset})$ where $\mathcal{B}_{i,p,j,\emptyset}$ is a sample set of size $b$. Therefore we have

$$\mathbb{E}[\exp cA_{j,s}] = \mathbb{E}[\exp c(\bar{v}_{p,j,\emptyset,k} - \nabla L_{t,k}(x_{i,p,j,\emptyset}))]$$

$$= \mathbb{E}\left[\exp c\left(\frac{1}{mb}\sum_{i=1}^{m}\sum_{s\in\mathcal{B}_{i,p,j,\emptyset}} \nabla l_k(x_{i,p,j,\emptyset}; s) - \nabla L_{t,k}(x_{i,p,j,\emptyset})\right)\right]$$

$$= \prod_{s\in\mathcal{B}_{i,p,j,\emptyset}}\prod_{i\in[m]} \mathbb{E}\left[\exp\frac{c}{bm}(\nabla l_k(x_{i,p,j,\emptyset}; s) - \nabla L_{t,k}(x_{i,p,j,\emptyset}))\right]$$

$$\leq \exp\left(\frac{c^2\alpha^2}{2bm}\right),$$

where the last inequality holds because for a random variable $X \in [-\alpha, \alpha]$, we have $\mathbb{E}[\exp c(X - \mathbb{E}[X])] \leq \exp\left(\frac{c^2\alpha^2}{2}\right)$.

Assume the depth of the vertex $|s| \geq 0$ and let $s = s'a$ where $a \in \{0,1\}$. If $a = 0$, clearly the lemma holds. If $a = 1$, recall that $v_{i,p,j,s} = v_{i,p,j,s'} + \nabla l(x_{i,p,j,s}; \mathcal{B}_{i,p,j,s}) - \nabla l(x_{i,p,j,s'}; \mathcal{B}_{i,p,j,s})$, then

$$A_{j,s} = \bar{v}_{p,j,s,k} - \nabla L_{t,k}(x_{i,p,j,s})$$

$$= A_{j,s'} + \frac{1}{m}\sum_{i=1}^{m}\nabla l_k(x_{i,p,j,s}; \mathcal{B}_{i,p,j,s}) - \frac{1}{m}\sum_{i=1}^{m}\nabla l_k(x_{i,p,j,s'}; \mathcal{B}_{i,p,j,s})$$

$$- \nabla L_{t,k}(x_{i,p,j,s}) + \nabla L_{t,k}(x_{i,p,j,s'})$$

Let $\mathcal{B}_{i,p,<(j,s)} = \cup_{(j_1,s_1)<(j,s)}\mathcal{B}_{i,p,j_1,s_1}$ be the set containing all the samples used up to vertex $(j,s)$ in phase $p$ at client $i$. We have

$$\mathbb{E}[\exp cA_{j,s}]$$

$$= \mathbb{E}\left[\exp\left(c(A_{j,s'} + \frac{1}{m}\sum_{i=1}^{m}\nabla l_k(x_{i,p,j,s}; \mathcal{B}_{i,p,j,s}) - \frac{1}{m}\sum_{i=1}^{m}\nabla l_k(x_{i,p,j,s'}; \mathcal{B}_{i,p,j,s})\right)\right.$$

$$\times \exp\left(-\nabla L_{t,k}(x_{i,p,j,s}) + \nabla L_{t,k}(x_{i,p,j,s'})\right)\bigg]$$

$$= \mathbb{E}\left[\mathbb{E}\left[\exp\left(c(A_{j,s'} + \frac{1}{m}\sum_{i=1}^{m}\nabla l_k(x_{i,p,j,s}; \mathcal{B}_{i,p,j,s}) - \frac{1}{m}\sum_{i=1}^{m}\nabla l_k(x_{i,p,j,s'}; \mathcal{B}_{i,p,j,s})\right.\right.$$

$$- \nabla L_{t,k}(x_{i,p,j,s}) + \nabla L_{t,k}(x_{i,p,j,s'}) \Big) \Big| \mathcal{B}_{i,p,<(j,s)} \Big] \Big]$$

$$= \mathbb{E} \Big[ \mathbb{E} \big[ \exp \left( c A_{j,s'} \right) | \mathcal{B}_{i,p,<(j,s)} \big]$$

$$\times \mathbb{E}[\exp \Big( c(\frac{1}{m} \sum_{i=1}^{m} \nabla l_k(x_{i,p,j,s}; \mathcal{B}_{i,p,j,s}) - \frac{1}{m} \sum_{i=1}^{m} \nabla l_k(x_{i,p,j,s'}; \mathcal{B}_{i,p,j,s})$$

$$- \nabla L_{t,k}(x_{i,p,j,s}) + \nabla L_{t,k}(x_{i,p,j,s'}) \Big) \Big| \mathcal{B}_{i,p,<(j,s)} \Big] \Big].$$

Since $l_{i,t}(\cdot; s)$ is $\beta$-smooth w.r.t. $\| \cdot \|_1$, we have

$$|\nabla l_k(x_{i,p,j,s}; \mathcal{B}_{i,p,j,s}) - \nabla l_k(x_{i,p,j,s'}; \mathcal{B}_{i,p,j,s})| \le \beta \|x_{i,p,j,s} - x_{i,p,j,s'}\|_1.$$

Since vertex $(j, s)$ is the right son of vertex $(j, s')$, the number of updates between $x_{i,p,j,s}$ and $x_{i,p,j,s'}$ is at most the number of leafs visited between these two vertices i.e. $2^{j-|s|}$. Therefore we have

$$\|x_{i,p,j,s} - x_{i,p,j,s'}\|_1 \le \eta_{i,p,j,s'} 2^{j-|s|} \le 2^{2-|s|}.$$

By using similar arguments to the case $|s| = 0$, we can get

$$\mathbb{E} \Big[ \exp \Big( c(\frac{1}{m} \sum_{i=1}^{m} \nabla l_k(x_{i,p,j,s}; \mathcal{B}_{i,p,j,s}) - \frac{1}{m} \sum_{i=1}^{m} \nabla l_k(x_{i,p,j,s'}; \mathcal{B}_{i,p,j,s}) \Big)$$

$$\times \exp \Big( - \nabla L_{t,k}(x_{i,p,j,s}) + \nabla L_{t,k}(x_{i,p,j,s'}) \Big) \Big| \mathcal{B}_{i,p,<(j,s)} \Big]$$

$$\le \exp \Big( \frac{O(1)c^2 \beta^2 2^{-2|s|}}{m|\mathcal{B}_{i,p,j,s}|} \Big)$$

$$\le \exp \Big( \frac{O(1)c^2 \beta^2 2^{-|s|}}{bm} \Big).$$

Then we get

$$\mathbb{E}[\exp c A_{j,s}] \le \mathbb{E}[\exp c A_{j,s'}] \exp \Big( \frac{O(1)c^2 \beta^2 2^{-|s|}}{bm} \Big).$$

Applying this inductively, we have for every index $1 \le k \le d$

$$\mathbb{E}[\exp c A_{j,s}] \le \exp \Big( \frac{O(1)c^2(\alpha^2 + \beta^2)}{bm} \Big).$$

$\square$

Lemma 3 upper bounds the variance of the average gradient.

**Lemma 3.** *At phase $p$, for each vertex $(j, s)$ and $2^{p-1} \le t < 2^p - 1$, we have*

$$\mathbb{E}\left[\|\bar{v}_{p,j,s} - \nabla L_t(x_{i,p,j,s})\|_\infty\right] \le (\alpha + \beta)O\left(\sqrt{\frac{\log d}{bm}}\right),$$

*where $\bar{v}_{p,j,s} = \frac{1}{m} \sum_{i=1}^{m} v_{i,p,j,s}$.*

*Proof.* Lemma 2 implies that $\bar{v}_{p,j,s,k} - \nabla L_{t,k}(x_{i,p,j,s})$ is $O(\frac{\alpha^2 + \beta^2}{bm})$-sub-Gaussian for every index $1 \le k \le d$ of the vectors. Applying standard results of the maximum of $d$ sub-Gaussian random variables, we get

$$\mathbb{E}[\|\bar{v}_{p,j,s} - \nabla L_t(x_{i,p,j,s})\|_\infty] \le O\left(\sqrt{\frac{\alpha^2 + \beta^2}{bm}}\right)\sqrt{\log d}.$$

$\square$

Lemma 4 gives the tail bound of the sum of i.i.d. random variables following Laplace distribution.

**Lemma 4.** *Let $\xi_{i,n}$ be IID random variables following the distribution $Lap(\lambda_{i,j,s})$. Then we have*

$$\mathbb{E}\left[\max_{n:1\leq n\leq d}\left|\sum_{i=1}^{m}\xi_{i,n}\right|\right] \leq O(\sqrt{m}\lambda_{i,j,s}\ln d).$$

*Proof.* $\xi_{i,n}$'s are $md$ IID random variables following the distribution $Lap(\lambda_{i,j,s})$. We note that

$$\mathbb{E}\left[\exp(u\xi_{i,n})\right] = \frac{1}{1-\lambda_{i,j,s}^2 u^2}, |u| \leq \frac{1}{\lambda_{i,j,s}}.$$

Since $\frac{1}{1-\lambda_{i,j,s}^2 u^2} \leq 1 + 2\lambda_{i,j,s}^2 u^2 \leq \exp(2\lambda_{i,j,s}^2 u^2)$, when $|u| \leq \frac{1}{2\lambda_{i,j,s}}$, $\xi_{i,n}$ is sub-exponential with parameter $(4\lambda_{i,j,s}^2, 2\lambda_{i,j,s})$. Applying standard results of linear combination of sub-exponential random variables, we can conclude that $\sum_{i=1}^{m}\xi_{i,n}$, denoted as $Y_n$, is sub-exponential with parameter $(4m\lambda_{i,j,s}^2, 2\lambda_{i,j,s})$. From standard results of tail bounds of sub-exponential random variables, we have

$$\mathbb{P}\left(|Y_n| \geq c\right) \leq 2\exp\left(-\frac{c^2}{8m\lambda_{i,j,s}^2}\right), \text{ if } 0 \leq c \leq 2m\lambda_{i,j,s},$$

$$\mathbb{P}(|Y_n| \geq c) \leq 2\exp\left(-\frac{c}{4\lambda_{i,j,s}}\right), \text{ if } c \geq 2m\lambda_{i,j,s}.$$

Since $\mathbb{P}\left(\max_{n:1\leq n\leq d}|Y_n| \geq c\right) \leq \sum_{n:1\leq n\leq d}\mathbb{P}\left(|Y_n| \geq c\right)$, we have

$$\mathbb{P}(\max_{n:1\leq n\leq d}|Y_n| \geq c) \leq 2d\exp\left(-\frac{c^2}{8m\lambda_{i,j,s}^2}\right), \text{ if } 0 \leq c \leq 2m\lambda_{i,j,s},$$

$$\mathbb{P}(\max_{n:1\leq n\leq d}|Y_n| \geq c) \leq 2d\exp\left(-\frac{c}{4\lambda_{i,j,s}}\right), \text{ if } c \geq 2m\lambda_{i,j,s}.$$

Then we have

$$\begin{aligned}
\mathbb{E}\left[\max_{n:1\leq n\leq d}|Y_n|\right] &= \int_0^\infty \mathbb{P}\left(\max_{n:1\leq n\leq d}|Y_n| \geq c\right) dc \\
&= \sqrt{8m\ln d}\lambda_{i,j,s} + \int_{\sqrt{8m\ln d}\lambda_{i,j,s}}^{2m\lambda_{i,j,s}} \mathbb{P}\left(\max_{n:1\leq n\leq d}|Y_n| \geq c\right) dc \\
&\quad + \int_{2m\lambda_{i,j,s}}^\infty \mathbb{P}\left(\max_{n:1\leq n\leq d}|Y_n| \geq c\right) dc \\
&\leq \sqrt{8m\ln d}\lambda_{i,j,s} + \int_{\sqrt{8m\ln d}\lambda_{i,j,s}}^{2m\lambda_{i,j,s}} 2\exp\left(\ln d - \frac{c^2}{8m\lambda_{i,j,s}^2}\right) dc \\
&\quad + \int_{2m\lambda_{i,j,s}}^\infty 2\exp\left(\ln d - \frac{c}{4\lambda_{i,j,s}}\right) dc \\
&\leq \sqrt{8m\ln d}\lambda_{i,j,s} + \sqrt{8m}\lambda_{i,j,s}\int_{-\infty}^{+\infty}\exp\left(-u^2\right) du \\
&\quad + 8\lambda_{i,j,s}\int_{2/m-\ln d}^\infty \exp(-u)du \\
&\leq \sqrt{8m\ln d}\lambda_{i,j,s} + \sqrt{8m}\lambda_{i,j,s}\int_{-\infty}^{+\infty}\exp\left(-u^2\right) du \\
&\quad + 8\lambda_{i,j,s}\ln d + 8\lambda_{i,j,s}\int_0^\infty \exp(-u)du \\
&= \sqrt{8m\ln d}\lambda_{i,j,s} + \sqrt{8m\pi}\lambda_{i,j,s} + 8\lambda_{i,j,s}\ln d + 8\lambda_{i,j,s} \\
&= O(\sqrt{m}\lambda_{i,j,s}\ln d).
\end{aligned}$$

$\square$

**Lemma 5.** *Setting $\lambda_{i,j,s} = \frac{4\alpha 2^j}{b\varepsilon}$, we have*

$$\mathbb{E}[\langle \bar{v}_{p,k}, \bar{w}_{p,k}\rangle] \le \mathbb{E}\left[\min_{w\in\mathcal{X}}\langle \bar{v}_{p,k}, w\rangle\right] + O\left(\frac{\alpha 2^j \ln d}{b\varepsilon\sqrt{m}}\right).$$

*Proof.* Since $\bar{w}_{p,k} = \arg\min_{c_n:1\le n\le d}\left[\frac{1}{m}\sum_{i=1}^m \left(\langle c_n, v_{i,p,k}\rangle + \xi_{i,n}\right)\right]$, where $\xi_{i,n} \sim \mathrm{Lap}(\lambda_{i,j,s})$, we denote $\bar{w}_{p,k}$ as $c_{n^\star}$ and we have

$$
\begin{aligned}
\langle \bar{w}_{p,k}, \bar{v}_{p,k}\rangle &= \langle c_{n^\star}, \bar{v}_{p,k}\rangle \\
&= \min_{n:1\le n\le d}\left(\langle c_n, \bar{v}_{p,k}\rangle + \frac{1}{m}\sum_{i=1}^m \xi_{i,n}\right) - \frac{1}{m}\sum_{i=1}^m \xi_{i,n^\star} \\
&\le \min_{n:1\le n\le d}\langle c_n, \bar{v}_{p,k}\rangle + \frac{1}{m}\max_{n:1\le n\le d}\sum_{i=1}^m \xi_{i,n} - \frac{1}{m}\min_{n:1\le n\le d}\sum_{i=1}^m \xi_{i,n} \\
&\le \min_{n:1\le n\le d}\langle c_n, \bar{v}_{p,k}\rangle + \frac{2}{m}\max_{n:1\le n\le d}\left|\sum_{i=1}^m \xi_{i,n}\right|.
\end{aligned}
$$

Applying Lemma 4, we get

$$\mathbb{E}[\langle \bar{v}_{p,k}, \bar{w}_{p,k}\rangle] \le \mathbb{E}\left[\min_{w\in\mathcal{X}}\langle \bar{v}_{p,k}, w\rangle\right] + O\left(\frac{\lambda_{i,j,s} \ln d}{\sqrt{m}}\right).$$

$\square$

With the lemmas above, we are ready to prove Theorem 1.

*Proof.* Lemma 1 implies the claim about privacy. We proceed to prove the regret upper bound.

$$
\begin{aligned}
L_t(x_{i,p,k+1}) &\overset{(a)}{\le} L_t(x_{i,p,k}) + \langle \nabla L_t(x_{i,p,k}), x_{i,p,k+1} - x_{i,p,k}\rangle + \beta\frac{\|x_{i,p,k+1} - x_{i,p,k}\|_1^2}{2} \\
&\overset{(b)}{\le} L_t(x_{i,p,k}) + \eta_{i,p,k}\langle \nabla L_t(x_{i,p,k}), \bar{w}_{p,k} - x_{i,p,k}\rangle + \frac{1}{2}\beta\eta_{i,p,k}{}^2 \\
&= L_t(x_{i,p,k}) + \eta_{i,p,k}\langle \nabla L_t(x_{i,p,k}), x^\star - x_{i,p,k}\rangle + \eta_{i,p,k}\langle \nabla L_t(x_{i,p,k}) - \bar{v}_{p,k}, \bar{w}_{p,k} - x^\star\rangle \\
&\quad + \eta_{i,p,k}\langle \bar{v}_{p,k}, \bar{w}_{p,k} - x^\star\rangle + \frac{1}{2}\beta\eta_{i,p,k}{}^2 \\
&\overset{(c)}{\le} L_t(x_{i,p,k}) + \eta_{i,p,k}[L_t(x^\star) - L_t(x_{i,p,k})] + \eta_{i,p,k}\|\nabla L_t(x_{i,p,k}) - \bar{v}_{p,k}\|_\infty + \eta_{i,p,k}(\langle \bar{v}_{p,k}, \bar{w}_{p,k}\rangle \\
&\quad - \min_{w\in\mathcal{X}}\langle \bar{v}_{p,k}, w\rangle) + \frac{1}{2}\beta\eta_{i,p,k}{}^2,
\end{aligned}
$$

where $(a)$ is due to $\beta$-smoothness, $(b)$ is because of the updating rule of $x_{i,p,k}$, and $(c)$ follows from the convexity of the loss function and Hölder's inequality.

Subtracting $L_t(x^\star)$ from each side and taking expectations, we have

$$
\begin{aligned}
\mathbb{E}[L_t(x_{i,p,k+1}) - L_t(x^\star)] &\le (1 - \eta_{i,p,k})\mathbb{E}[L_t(x_{i,p,k}) - L_t(x^\star)] + \eta_{i,p,k}\mathbb{E}[\|\nabla L_t(x_{i,p,k}) - \bar{v}_{p,k}\|_\infty] \\
&\quad + \eta_{i,p,k}\mathbb{E}[\langle \bar{v}_{p,k}, \bar{w}_{p,k}\rangle - \min_{w\in\mathcal{X}}\langle \bar{v}_{p,k}, w\rangle] + \frac{1}{2}\beta\eta_{i,p,k}{}^2.
\end{aligned}
$$

Applying Lemma 5 and Lemma 3, we have

$$
\begin{aligned}
\mathbb{E}[L_t(x_{i,p,k+1}) - L_t(x^\star)] &\le (1 - \eta_{i,p,k})\mathbb{E}[L_t(x_{i,p,k}) - L_t(x^\star)] + \eta_{i,p,k}(\alpha + \beta)O\left(\sqrt{\frac{\log d}{bm}}\right) \\
&\quad + \frac{\eta_{i,p,k}\alpha 2^j}{b\varepsilon}O\left(\frac{\ln d}{\sqrt{m}}\right) + \frac{1}{2}\beta\eta_{i,p,k}{}^2.
\end{aligned}
$$

Let $\alpha_k = \eta_{i,p,k}(\alpha + \beta)O\left(\sqrt{\frac{\log d}{bm}}\right) + \frac{\eta_{i,p,k}\alpha 2^j}{b\varepsilon}O\left(\frac{\ln d}{\sqrt{m}}\right) + \frac{1}{2}\beta\eta_{i,p,k}{}^2$. We simplify the notion of $\eta_{i,p,k}$ to $\eta_k$. Then we have

$$\mathbb{E}[L_t(x_{i,p,k}) - L_t(x^\star)] \leq \sum_{k'=1}^{K}\alpha_k\prod_{i>k}^{K-1}(1 - \eta_{k'})$$

$$= \sum_{k=1}^{K}\alpha_k\frac{(k+1)k}{K(K-1)}$$

$$\leq \sum_{k=1}^{K}\alpha_k\frac{(k+1)^2}{(K-1)^2},$$

where $\eta_k = \frac{2}{k+1}$.

Since $K = 2^{T_1}$, simply algebra implies that

$$\mathbb{E}[L_t(x_{i,p,K}) - L_t(x^\star)] \leq O\left((\alpha + \beta)\sqrt{\frac{\log d}{bm}} + \frac{\alpha 2^{T_1}\ln d}{b\varepsilon\sqrt{m}} + \frac{\beta}{2^{T_1}}\right). \tag{4}$$

At iteration $2^{p-1} \leq t < 2^p$, setting $b = \frac{2^{p-1}}{(p-1)^2}$ and $T_1 = \frac{1}{2}\log\left(\frac{b\varepsilon\beta\sqrt{m}}{\alpha\log d}\right)$ in Equation (4), we have

$$\mathbb{E}[L_t(x_{i,p,K}) - L_t(x^\star)] \leq O\left((\alpha + \beta)p\sqrt{\frac{\log d}{2^p m}} + \frac{p\sqrt{\alpha\beta}\log d}{m^{\frac{1}{4}}\sqrt{2^p\varepsilon}}\right).$$

Therefore, the total regret from time step $2^p$ to $2^{p+1} - 1$ is at most

$$\mathbb{E}\left[\sum_{t=2^p}^{2^{p+1}-1}L_t(x_{i,t}) - \min_{u\in\mathcal{X}}\sum_{t=2^p}^{2^{p+1}-1}L_t(u)\right] \leq O\left((\alpha + \beta)p\sqrt{\frac{2^p\log d}{m}} + \frac{p\sqrt{\alpha\beta}\log d\sqrt{2^p}}{m^{\frac{1}{4}}\sqrt{\varepsilon}}\right).$$

Summing over $p$, we can get

$$\mathbb{E}\left[\sum_{t=1}^{T}L_t(x_{i,t}) - \min_{u\in\mathcal{X}}\sum_{t=1}^{T}L_t(u)\right] \leq \sum_{p=1}^{\log T}O\left((\alpha + \beta)p\sqrt{\frac{2^p\log d}{m}} + \frac{p\sqrt{\alpha\beta}\log d\sqrt{2^p}}{m^{\frac{1}{4}}\sqrt{\varepsilon}}\right)$$

$$\leq O\left((\alpha + \beta)\sum_{p=1}^{\log T}p\sqrt{\frac{2^p\log d}{m}} + \frac{\sqrt{\alpha\beta}\log d}{m^{\frac{1}{4}}\sqrt{\varepsilon}}\sum_{p=1}^{\log T}p\sqrt{2^p}\right)$$

$$\leq O\left((\alpha + \beta)\log T\sqrt{\frac{T\log d}{m}} + \frac{\sqrt{\alpha\beta}\log d\sqrt{T}\log T}{m^{\frac{1}{4}}\sqrt{\varepsilon}}\right).$$

Now we turn our focus to communication cost. Since there are $\log T$ phases, and within each phase, there are $O(2^{T_1})$ leaf vertices where communication is initiated, the communication frequency scales in $O(\sum_p 2^{T_1})$. Therefore, the communication cost scales in $O(m^{5/4}d\sqrt{T\varepsilon\beta/(\alpha\log d)})$.    □

**Corollary 2** (Restatement of Corollary 1). *If $\beta = O(\frac{\log d}{\sqrt{mT\varepsilon}})$, then, Fed-DP-OPE-Stoch (i) satisfies $\varepsilon$-DP and (ii) achieves the per-client regret of*

$$Reg(T, m) = O\left(\sqrt{\frac{T\log d}{m}} + \frac{\log T\log d}{\sqrt{m}\varepsilon}\right),$$

*with (iii) a communication cost of $O\left(md\log T\right)$.*

*Proof.* Similar to the proof of Theorem 1, we can show that

$$\mathbb{E}[L_t(x_{i,p,K}) - L_t(x^\star)] \le O\left( (\alpha + \beta)\sqrt{\frac{\log d}{bm}} + \frac{\alpha 2^{T_1} \ln d}{b\varepsilon\sqrt{m}} + \frac{\beta}{2^{T_1}} \right). \tag{5}$$

At iteration $2^{p-1} \le t < 2^p$, setting $b = 2^{p-1}$ and $T_1 = 1$ in Equation (5), we have

$$\mathbb{E}[L_t(x_{i,p,K}) - L_t(x^\star)] \le O\left( (\alpha + \beta)\sqrt{\frac{\log d}{2^p m}} + \frac{\alpha \ln d}{2^p \varepsilon\sqrt{m}} + \beta \right).$$

Since $\beta = O(\frac{\log d}{\sqrt{mT}\varepsilon})$, we have

$$\mathbb{E}[L_t(x_{i,p,K}) - L_t(x^\star)] \le O\left( (\alpha + \beta)\sqrt{\frac{\log d}{2^p m}} + \frac{\alpha \ln d}{2^p \varepsilon\sqrt{m}} \right).$$

Therefore, the total regret from time step $2^p$ to $2^{p+1}$ is at most

$$\mathbb{E}\left[ \sum_{t=2^p}^{2^{p+1}-1} L_t(x_{i,t}) - \min_{u\in\mathcal{X}} \sum_{t=2^p}^{2^{p+1}-1} L_t(u) \right] \le O\left( \sqrt{\frac{2^p \log d}{m}} + \frac{\log d}{\sqrt{m}\varepsilon} \right).$$

Summing over $p$, we can get

$$\begin{aligned}
\mathbb{E}\left[ \sum_{t=1}^{T} L_t(x_{i,t}) - \min_{u\in\mathcal{X}} \sum_{t=1}^{T} L_t(u) \right] &\le \sum_{p=1}^{\log T} O\left( \sqrt{\frac{2^p \log d}{m}} + \frac{\alpha \log d}{\sqrt{m}\varepsilon} \right) \\
&\le O\left( \sum_{p=1}^{\log T} \sqrt{\frac{2^p \log d}{m}} + \frac{\log d \log T}{\sqrt{m}\varepsilon} \right) \\
&\le O\left( \sqrt{\frac{T \log d}{m}} + \frac{\log d \log T}{\sqrt{m}\varepsilon} \right).
\end{aligned}$$

The proof of the DP guarantee and communication costs follows from the proof of Theorem 1. $\quad\square$

### C.3 Theorem 6 (for approximate DP)

**Theorem 6.** *Let $\delta \le 1/T$. Assume that loss function $l_{i,t}(\cdot)$ is convex, $\alpha$-Lipschitz, $\beta$-smooth w.r.t. $\|\cdot\|_1$. Assume $\varepsilon \le \frac{(\alpha \log(2^{p-1}/\delta) \log d)^{1/4}\sqrt{p-1}}{(\beta 2^{p-1})^{1/4}}$. Setting $\lambda_{i,j,s} = \frac{\alpha 2^{T_1/2} \log(2^{p-1}/\delta)}{b\varepsilon}$, $b = \frac{2^{p-1}}{(p-1)^2}$, and $T_1 = \frac{2}{3}\log\left( \frac{b\varepsilon\sqrt{m}\beta}{\alpha \log(2^{p-1}/\delta) \log d} \right)$, Fed-DP-OPE-Stoch is $(\varepsilon, \delta)$-DP, the communication cost scales in $O\left(md2^{T_1}\log T\right)$ and the per-client regret is upper bounded by*

$$O\left( (\alpha + \beta)\log T\sqrt{\frac{T \log d}{m}} + \frac{\alpha^{2/3}\beta^{1/3}\log^{2/3}(d)\log^{2/3}(1/\delta)T^{1/3}\log^{4/3}(T)}{m^{1/3}\varepsilon^{2/3}} \right).$$

*Proof.* We apply the following lemma to prove privacy in this setting.

**Lemma 6** (Asi et al. (2021b), Lemma 4.4). *Let $b \ge 2^{T_1}$, $\delta \le 1/T$ and $\varepsilon \le \sqrt{2^{-T_1}\log(1/\delta)}$. Setting $\lambda_{i,j,s} = \frac{\alpha 2^{T_1/2}\log(2^{p-1}/\delta)}{b\varepsilon}$, we have Fed-DP-OPE is $(O(\varepsilon), O(\delta))$-DP.*

Theorem 6 then follows using similar arguments to the proof of Theorem 1. $\quad\square$

## C.4 Extension to Central DP setting

In this section, we extend our Fed-DP-OPE-Stoch algorithm to the setting where client-server communication is secure with some modifications, achieving better utility performance.

We present the algorithm design in Algorithm 4 and Algorithm 5. We change the local privatization process to a global privatization process on the server side. Specifically, in Line 8, when communication is triggered, each client sends $\{\langle c_n, v_{i,j,s}\rangle\}_{n\in[d]}$ to the server, where $c_1, \ldots, c_d$ represents $d$ vertices of decision set $\mathcal{X} = \Delta_d$. In Line 5, after receiving $\{\langle c_n, v_{i,j,s}\rangle\}_{n\in[d]}$ from all clients, the central server privately predicts a new expert:

$$\bar{w}_{j,s} = \underset{c_n:1\leq n\leq d}{\arg\min}\left[\frac{1}{m}\sum_{i=1}^{m}\langle c_n, v_{i,j,s}\rangle + \zeta_n\right], \tag{6}$$

where $\zeta_n \sim \mathrm{Lap}(\mu_{j,s})$ and $\mu_{j,s} = \frac{4\alpha 2^j}{bm\varepsilon}$. Other components remain the same.

---

**Algorithm 4** Fed-DP-OPE-Stoch (CDP): Client $i$

---

1: **Input:** Phases $P$, trees $T_1$, decision set $\mathcal{X} = \Delta_d$ with vertices $\{c_1, \ldots, c_d\}$, batch size $b$.
2: **Initialize:** Set $x_{i,1} = z \in \mathcal{X}$ and pay cost $l_{i,1}(x_{i,1})$.
3: **for** $p = 2$ to $P$ **do**
4:     Set $\mathcal{B}_{i,p} = \{l_{i,2^{p-2}}, \ldots, l_{i,2^{p-1}-1}\}$
5:     Set $k = 1$ and $x_{i,p,1} = x_{i,p-1,K}$
6:     $\{v_{i,j,s}\}_{j\in[T_1],s\in\{0,1\}^{\leq j}} = $ DP-FW $(\mathcal{B}_{i,p}, T_1, b)$
7:     **for all** leaf vertices $s$ reached in DP-FW **do**
8:         **Communicate to server:** $\{\langle c_n, v_{i,j,s}\rangle\}_{n\in[d]}$
9:         **Receive from server:** $\bar{w}_{j,s}$
10:        Update $x_{i,p,k}$ according to Equation (2)
11:        Update $k = k + 1$
12:     **end for**
13:     Final iterate outputs $x_{i,p,K}$
14:     **for** $t = 2^{p-1}$ to $2^p - 1$ **do**
15:        Receive loss $l_{i,t} : \mathcal{X} \rightarrow \mathbb{R}$ and pay cost $l_{i,t}(x_{i,p,K})$
16:     **end for**
17: **end for**

---

---

**Algorithm 5** Fed-DP-OPE-Stoch (CDP): Central server

---

1: **Input:** Phases $P$, number of clients $m$, decision set $\mathcal{X} = \Delta_d$ with vertices $\{c_1, \ldots, c_d\}$.
2: **Initialize:** Pick any $z \in \mathcal{X}$ and broadcast to clients.
3: **for** $p = 2$ to $P$ **do**
4:     **Receive from clients:** $\{\langle c_n, v_{i,j,s}\rangle\}_{n\in[d]}$
5:     $\bar{w}_{j,s} = \underset{c_n:1\leq n\leq d}{\arg\min}\left[\frac{1}{m}\sum_{i=1}^{m}\langle c_n, v_{i,j,s}\rangle + \zeta_n\right]$
6:     **Communicate to clients:** $\bar{w}_{j,s}$
7: **end for**

---

Now we present the theoretical guarantees in this setting.

**Theorem 7.** *Assume that loss function $l_{i,t}(\cdot)$ is convex, $\alpha$-Lipschitz, $\beta$-smooth w.r.t. $\|\cdot\|_1$. Fed-DP-OPE-Stoch (i) satisfies $\varepsilon$-DP and (ii) achieves the per-client regret of*

$$O\left((\alpha + \beta)\log T\sqrt{\frac{T\log d}{m}} + \frac{\sqrt{\alpha\beta}\log d\sqrt{T}\log T}{\sqrt{m}\sqrt{\varepsilon}}\right),$$

*with (iii) a communication cost of $O\left(m^{\frac{3}{2}}d\sqrt{\frac{T\varepsilon\beta}{\alpha\log d}}\right)$.*

To prove the theorem, we start with Lemma 7 that gives pure privacy guarantees.

**Lemma 7.** *Assume that $2^{T_1} \leq b$. Setting $\mu_{j,s} = \frac{4\alpha 2^j}{bm\varepsilon}$, Fed-DP-OPE-Stoch is $\varepsilon$-DP.*

*Proof.* Let $\mathcal{S}$ and $\mathcal{S}'$ be two neighboring datasets and assume that they differ in sample $l_{i_1,t_1}$ or $l'_{i_1,t_1}$, where $2^{p_1-1} \le t_1 < 2^{p_1}$. The algorithm is $\varepsilon$-DP if we have $(x_{1,1,K}, \ldots, x_{m,P,K}) \approx_{(\varepsilon,0)} (x'_{1,1,K}, \ldots, x'_{m,P,K})$.

Let $\mathcal{B}_{i_1,j,s}$ be the set that contains $l_{i_1,t_1}$ or $l'_{i_1,t_1}$. Recall that $|\mathcal{B}_{i_1,j,s}| = 2^{-|s|}b$. The key point is that this set is used in the calculation of $v_{i_1,p_1,k}$ for at most $2^{j-|s|}$ iterates, i.e. the leaves that are descendants of the vertex. Let $k_0$ and $k_1$ be the first and last iterate such that $\mathcal{B}_{i_1,j,s}$ is used for the calculation of $v_{i_1,j,k}$, hence $k_1 - k_0 + 1 \le 2^{j-|s|}$. For a sequence $a_i, \ldots, a_j$, we use the shorthand $a_{i:j} = \{a_i, \ldots, a_j\}$.

**Step 1: $\mathbf{x_{1:m,p_1,k_0:k_1}} \approx_{(\varepsilon,0)} \mathbf{x'_{1:m,p_1,k_0:k_1}}$ and $\mathbf{\bar{w}_{p_1,k_0:k_1}} \approx_{(\varepsilon,0)} \mathbf{\bar{w}'_{p_1,k_0:k_1}}$ by basic composition, post-processing and report noisy max**

We will show that $(x_{1,p_1,k_0}, \ldots, x_{m,p_1,k_1})$ and $(x'_{1,p_1,k_0}, \ldots, x'_{m,p_1,k_1})$ are $\varepsilon$-indistinguishable. Since $\mathcal{B}_{i_1,j,s}$ is used to calculate $v_{i_1,p_1,k}$ for at most $2^{j-|s|}$ iterates, i.e. $k_1 - k_0 + 1 \le 2^{j-|s|}$, it is enough to show that $\bar{w}_{p_1,k} \approx_{(\frac{\varepsilon}{2^{j-|s|}},0)} \bar{w}'_{p_1,k}$ for $k_0 \le k \le k_1$ and then apply basic composition and post-processing. Note that for every $k_0 \le k \le k_1$, the sensitivity $|\langle c_n, v_{i_1,p_1,k} - v'_{i_1,p_1,k}\rangle| \le \frac{4\alpha}{2^{-|s|}b}$, therefore, $|\frac{1}{m}\sum_{i=1}^m \langle c_n, v_{i,p_1,k}\rangle - \frac{1}{m}\sum_{i=1}^m \langle c_n, v'_{i,p_1,k}\rangle| \le \frac{4\alpha}{2^{-|s|}mb}$. Using privacy guarantees of report noisy max (Lemma 15), we have $\bar{w}_{p_1,k} \approx_{(\frac{\varepsilon}{2^{j-|s|}},0)} \bar{w}'_{p_1,k}$ for $k_0 \le k \le k_1$ with $\mu_{j,s} = \frac{4\alpha 2^j}{bm\varepsilon}$.

**Step 2: $\mathbf{x_{1:m,1:P,K}} \approx_{(\varepsilon,0)} \mathbf{x'_{1:m,1:P,K}}$ by post-processing**

In order to show $(x_{1,1,K}, \ldots, x_{m,P,K}) \approx_{(\varepsilon,0)} (x'_{1,1,K}, \ldots, x'_{m,P,K})$, we only need to prove that $(x_{1,p_1,K}, \ldots, x_{m,p_1,K}) \approx_{(\varepsilon,0)} (x'_{1,p_1,K}, \ldots, x'_{m,p_1,K})$ and apply post-processing. It is enough to show that iterates $(x_{1,p_1,1}, \ldots, x_{m,p_1,K})$ and $(x'_{1,p_1,1}, \ldots, x'_{m,p_1,K})$ is $\varepsilon$-indistinguishable.

The iterates $x_{1:m,p_1,1:k_0-1}$ and $x'_{1:m,p_1,1:k_0-1}$ do not depend on $l_{i_1,t_1}$ or $l'_{i_1,t_1}$, hence 0-indistinguishable. Moreover, given that $(x_{1,p_1,k_0}, \ldots, x_{m,p_1,k_1})$ and $(x'_{1,p_1,k_0}, \ldots, x'_{m,p_1,k_1})$ are $\varepsilon$-indistinguishable, it is clear that $(x_{1,p_1,k_1+1}, \ldots, x_{m,p_1,K})$ and $(x'_{1,p_1,k_1+1}, \ldots, x'_{m,p_1,K})$ are $\varepsilon$-indistinguishable by post-processing. $\square$

To help prove the upper bound of the regret, we introduce some lemmas first.

**Lemma 8.** *Setting $\mu_{j,s} = \frac{4\alpha 2^j}{mb\varepsilon}$, we have*

$$\mathbb{E}[\langle \bar{v}_{p,k}, \bar{w}_{p,k}\rangle] \le \mathbb{E}\left[\min_{w\in\mathcal{X}}\langle \bar{v}_{p,k}, w\rangle\right] + O\left(\frac{\alpha 2^j \ln d}{mb\varepsilon}\right).$$

*Proof.* Since $\bar{w}_{p,k} = \arg\min_{c_n:1\le n\le d}\left[\frac{1}{m}\sum_{i=1}^m \langle c_n, v_{i,p,k}\rangle + \zeta_n\right]$, where $\zeta_n \sim \mathrm{Lap}(\mu_{j,s})$, we denote $\bar{w}_{p,k}$ as $c_{n^\star}$ and we have

$$
\begin{aligned}
\langle \bar{w}_{p,k}, \bar{v}_{p,k}\rangle &= \langle c_{n^\star}, \bar{v}_{p,k}\rangle \\
&= \min_{n:1\le n\le d}\left(\langle c_n, \bar{v}_{p,k}\rangle + \zeta_n\right) - \zeta_{n^\star} \\
&\le \min_{n:1\le n\le d}\langle c_n, \bar{v}_{p,k}\rangle + 2\max_{n:1\le n\le d}|\zeta_n|.
\end{aligned}
$$

Standard results for the expectation of maximum of $d$ i.i.d. Laplace random variables imply that $\mathbb{E}\left[\max_{n:1\le n\le d}|\zeta_n|\right] \le O(\mu_{j,s}\ln d)$. Therefore,

$$\mathbb{E}[\langle \bar{v}_{p,k}, \bar{w}_{p,k}\rangle] \le \mathbb{E}\left[\min_{w\in\mathcal{X}}\langle \bar{v}_{p,k}, w\rangle\right] + O\left(\mu_{j,s}\ln d\right).$$

$\square$

Then we are ready to prove Theorem 7.

*Proof.* Lemma 7 implies the claim about privacy. Following the same arguments in the proof of Theorem 1, we can get

$$\mathbb{E}[L_t(x_{i,p,k+1}) - L_t(x^\star)] \le (1 - \eta_{i,p,k})\mathbb{E}[L_t(x_{i,p,k}) - L_t(x^\star)] + \eta_{i,p,k}\mathbb{E}[\|\nabla L_t(x_{i,p,k}) - \bar{v}_{p,k}\|_\infty]$$

$$+ \eta_{i,p,k}\mathbb{E}[\langle \bar{v}_{p,k}, \bar{w}_{p,k}\rangle - \min_{w \in \mathcal{X}}\langle \bar{v}_{p,k}, w\rangle] + \frac{1}{2}\beta\eta_{i,p,k}{}^2.$$

Applying Lemma 8 and Lemma 3, we have

$$\mathbb{E}[L_t(x_{i,p,k+1}) - L_t(x^\star)] \le (1 - \eta_{i,p,k})\mathbb{E}[L_t(x_{i,p,k}) - L_t(x^\star)] + \eta_{i,p,k}(\alpha + \beta)O\left(\sqrt{\frac{\log d}{bm}}\right)$$

$$+ \frac{\eta_{i,p,k}\alpha 2^j}{b\varepsilon}O\left(\frac{\ln d}{m}\right) + \frac{1}{2}\beta\eta_{i,p,k}{}^2.$$

Let $\alpha_k = \eta_{i,p,k}(\alpha + \beta)O\left(\sqrt{\frac{\log d}{bm}}\right) + \frac{\eta_{i,p,k}\alpha 2^j}{b\varepsilon}O\left(\frac{\ln d}{m}\right) + \frac{1}{2}\beta\eta_{i,p,k}{}^2$. We simplify the notion of $\eta_{i,p,k}$ to $\eta_k$. Then we have

$$\mathbb{E}[L_t(x_{i,p,k}) - L_t(x^\star)] \le \sum_{k'=1}^{K}\alpha_k \prod_{i>k}^{K-1}(1 - \eta_{k'})$$

$$= \sum_{k=1}^{K}\alpha_k \frac{(k+1)k}{K(K-1)}$$

$$\le \sum_{k=1}^{K}\alpha_k \frac{(k+1)^2}{(K-1)^2},$$

where $\eta_k = \frac{2}{k+1}$.

Since $K = 2^{T_1}$, simply algebra implies that

$$\mathbb{E}[L_t(x_{i,p,K}) - L_t(x^\star)] \le O\left((\alpha + \beta)\sqrt{\frac{\log d}{bm}} + \frac{\alpha 2^{T_1}\ln d}{b\varepsilon m} + \frac{\beta}{2^{T_1}}\right). \qquad (7)$$

At iteration $2^{p-1} \le t < 2^p$, setting $b = \frac{2^{p-1}}{(p-1)^2}$ and $T_1 = \frac{1}{2}\log\left(\frac{b\varepsilon\beta m}{\alpha \log d}\right)$, we have

$$\mathbb{E}[L_t(x_{i,p,K}) - L_t(x^\star)] \le O\left((\alpha + \beta)p\sqrt{\frac{\log d}{2^p m}} + \frac{p\sqrt{\alpha\beta}\log d}{\sqrt{m}\sqrt{2^p \varepsilon}}\right).$$

Summing over all the timesteps, we have

$$\mathbb{E}\left[\sum_{t=1}^{T}L_t(x_{i,t}) - \min_{u \in \mathcal{X}}\sum_{t=1}^{T}L_t(u)\right] \le O\left((\alpha + \beta)\log T\sqrt{\frac{T\log d}{m}} + \frac{\sqrt{\alpha\beta}\log d\sqrt{T}\log T}{\sqrt{m}\varepsilon}\right).$$

The proof of communication cost is similar to that in proof of Theorem 1. $\qquad \square$

## D  Proof of Lower Bounds

**Theorem 8** (Restatement of Theorem 2). *For any federated OPE algorithm against oblivious adversaries, the per-client regret is lower bounded by* $\Omega(\sqrt{T\log d})$. *Let* $\varepsilon \in (0,1]$ *and* $\delta = o(1/T)$, *for any* $(\varepsilon, \delta)$-*DP federated OPE algorithm, the per-client regret is lower bounded by* $\Omega\left(\min\left(\frac{\log d}{\varepsilon}, T\right)\right)$.

*Proof.* Consider the case when all clients receive the same loss function from the oblivious adversary at each time step, i.e. $l_{i,t} = l'_t$. Then we define the average policy among all clients $p'_t(k) = \frac{1}{m}\sum_{i=1}^{m}p_{i,t}(k), \forall k \in [d]$. Now the regret is

$$\mathbb{E}\left[\frac{1}{m}\sum_{i=1}^{m}\sum_{t=1}^{T}l_{i,t}(x_{i,t})\right] - \frac{1}{m}\sum_{i=1}^{m}\sum_{t=1}^{T}l_{i,t}(x^\star) = \mathbb{E}\left[\frac{1}{m}\sum_{i=1}^{m}\sum_{t=1}^{T}l'_t(x_{i,t})\right] - \sum_{t=1}^{T}l'_t(x^\star)$$

$$= \frac{1}{m} \sum_{i=1}^{m} \sum_{t=1}^{T} \sum_{k=1}^{d} p_{i,t}(k) \cdot l'_t(k) - \sum_{t=1}^{T} l'_t(x^\star)$$

$$= \sum_{t=1}^{T} \sum_{k=1}^{d} \left( \frac{1}{m} \sum_{i=1}^{m} p_{i,t}(k) \right) \cdot l'_t(k) - \sum_{t=1}^{T} l'_t(x^\star)$$

$$= \sum_{t=1}^{T} \sum_{k=1}^{d} p'_t(k) \cdot l'_t(k) - \sum_{t=1}^{T} l'_t(x^\star).$$

Note that $p'_t(k)$ is defined by $p_{1,t}(k), \ldots, p_{m,t}(k)$, which in turn are determined by $l_{1,1}, \ldots, l_{m,t-1}$. According to our choice of $l_{i,t} = l'_t$, $p'_t(k)$ is determined by $l'_1, l'_2, \ldots, l'_{t-1}$. Therefore $p'_1, p'_2, \ldots, p'_t$ are generated by a legitimate algorithm for online learning with expert advice problems.

There exists a sequence of losses $l'_1, l'_2, \ldots, l'_t$ such that for any algorithm for online learning with expert advice problem, the expected regret satisfies (Cesa-Bianchi and Lugosi (2006), Theorem 3.7)

$$\sum_{t=1}^{T} \sum_{k=1}^{d} p'_t(k) \cdot l'_t(k) - \sum_{t=1}^{T} l'_t(x^\star) \geq \Omega(\sqrt{T \log d}).$$

Therefore, we have

$$\mathbb{E} \left[ \frac{1}{m} \sum_{i=1}^{m} \sum_{t=1}^{T} l_{i,t}(x_{i,t}) \right] - \frac{1}{m} \sum_{i=1}^{m} \sum_{t=1}^{T} l_{i,t}(x^\star) \geq \Omega(\sqrt{T \log d}).$$

From Lemma 9, if $\varepsilon \in (0, 1]$ and $\delta = o(1/T)$, then there exists a sequence of losses $l'_1, l'_2, \ldots, l'_t$ such that for any $(\varepsilon, \delta)$-DP algorithm for online learning with expert advice problem against oblivious adversaries, the expected regret satisfies

$$\sum_{t=1}^{T} \sum_{k=1}^{d} p'_t(k) \cdot l'_t(k) - \sum_{t=1}^{T} l'_t(x^\star) \geq \Omega \left( \min \left( \frac{\log d}{\varepsilon}, T \right) \right).$$

Therefore, we have for any $(\varepsilon, \delta)$-DP algorithm,

$$\mathbb{E} \left[ \frac{1}{m} \sum_{i=1}^{m} \sum_{t=1}^{T} l_{i,t}(x_{i,t}) \right] - \frac{1}{m} \sum_{i=1}^{m} \sum_{t=1}^{T} l_{i,t}(x^\star) \geq \Omega \left( \min \left( \frac{\log d}{\varepsilon}, T \right) \right).$$

$\square$

**Lemma 9.** *Let $\varepsilon \in (0, 1]$ and $\delta = o(1/T)$. There exists a sequence of losses $l_1, \ldots, l_T$ such that for any $(\varepsilon, \delta)$-DP algorithm $\mathcal{A}$ for DP-OPE against oblivious adversaries satisfies*

$$\sum_{t=1}^{T} l_t(x_t) - \min_{x^\star \in [d]} \sum_{t=1}^{T} l_t(x^\star) \geq \Omega \left( \min \left( \frac{\log d}{\varepsilon}, T \right) \right).$$

*Proof.* Let $n, d \in \mathbb{N}$. Define $\mathbf{y} \in \mathcal{Y}^n$ containing $n$ records, where $\mathcal{Y} = \{0, 1\}^d$. The function 1-Select$_d$: $\mathcal{Y}^n \to [d]$ corresponds to selecting a coordinate $b \in [d]$ in the batch model.

Then we define the regret of 1-Select$_d$. For a batched algorithm $\mathcal{M}$ with input dataset $\mathbf{y}$ and output $b \in [d]$, define

$$\text{Reg}_{1\text{-Select}_d}(\mathcal{M}(\mathbf{y})) = \frac{1}{n} \left[ \sum_{i=1}^{n} y_i(b) - \min_{x^\star \in [d]} \left( \sum_{i=1}^{n} y_i(x^\star) \right) \right].$$

Let $\mathcal{A}$ be an $(\varepsilon, \delta)$-DP algorithm $\left( \{0, 1\}^d \right)^T \to ([d])^T$ for DP-OPE against oblivious adversaries with regret $\sum_{t=1}^{T} l_t(x_t) - \min_{x^\star \in [d]} \sum_{t=1}^{T} l_t(x^\star) \leq \alpha$. Setting $T = n$, we can use $\mathcal{A}$ to construct

an $(\varepsilon, \delta)$-DP algorithm $\mathcal{M}$ for 1-Select$_d$ in the batch model. The details of the algorithm appear in Algorithm 6.

---

**Algorithm 6** Batch algorithm $\mathcal{M}$ for 1-Select (Jain et al. (2023), Algorithm 2 with $k = 1$)

---

1: **Input:** $\mathbf{y} = (y_1, \ldots, y_n) \in \mathcal{Y}^n$, where $\mathcal{Y} = \{0, 1\}^d$, and black-box access to a DP-OPE algorithm $\mathcal{A}$ for oblivious adversaries.
2: Construct a stream $\mathbf{z} \leftarrow \mathbf{y}$ with $n$ records.
3: **for** $t = 1$ to $n$ **do**
4:     Send the record $z_t$ to $\mathcal{A}$ and get the corresponding output $x_t$.
5: **end for**
6: Output $b = x_n$.

---

Let $\varepsilon > 0, \alpha \in \mathbb{R}^+$, and $T, d, n \in \mathbb{N}$, where $T = n$. If a DP-OPE algorithm $\mathcal{A}: \left(\{0, 1\}^d\right)^T \to ([d])^T$ for oblivious adversaries is $(\varepsilon, \delta)$-DP and the regret is upper bounded by $\alpha$, i.e. $\sum_{t=1}^{T} l_t(x_t) - \min_{x^\star \in [d]} \sum_{t=1}^{T} l_t(x^\star) \leq \alpha$, then by Lemma 10, we have the batch algorithm $\mathcal{M}$ for 1-Select$_d$ is $(\varepsilon, \delta)$-DP and $\mathrm{Reg}_{1\text{-Select}_d}(\mathcal{M}) \leq \frac{\alpha}{n}$.

If $\delta = o(1/T)$, then $n = \Omega\left(\frac{n \log d}{\varepsilon \alpha}\right)$ (Lemma 11). We have $\alpha = \min\left(\Omega\left(\frac{\log d}{\varepsilon}\right), n\right) = \min\left(\Omega\left(\frac{\log d}{\varepsilon}\right), T\right)$. So $\alpha \geq \Omega\left(\min\left(\frac{\log d}{\varepsilon}, T\right)\right)$. Therefore, if an algorithm for DP-OPE against oblivious adversaries is $(\varepsilon, \delta)$-differentially private and $\sum_{t=1}^{T} l_t(x_t) - \min_{x^\star \in [d]} \sum_{t=1}^{T} l_t(x^\star) \leq \alpha$ holds, then $\alpha \geq \Omega\left(\min\left(\frac{\log d}{\varepsilon}, T\right)\right)$. This means that there exists a sequence of loss functions $l_1, \ldots, l_T$ such that $\sum_{t=1}^{T} l_t(x_t) - \min_{x^\star \in [d]} \sum_{t=1}^{T} l_t(x^\star) \geq \Omega\left(\min\left(\frac{\log d}{\varepsilon}, T\right)\right)$. $\qquad\square$

**Lemma 10.** *Let $\mathcal{M}$ be the batch algorithm for 1-Select$_d$. For all $\varepsilon > 0, \delta \geq 0, \alpha \in \mathbb{R}^+$, and $T, d, n \in \mathbb{N}$, where $T = n$, if a DP-OPE algorithm $\mathcal{A}: \left(\{0, 1\}^d\right)^T \to ([d])^T$ for oblivious adversaries is $(\varepsilon, \delta)$-differentially private and the regret is upper bounded by $\alpha$, i.e. $\sum_{t=1}^{T} l_t(x_t) - \min_{x^\star \in [d]} \sum_{t=1}^{T} l_t(x^\star) \leq \alpha$, then batch algorithm $\mathcal{M}$ for 1-Select$_d$ is $(\varepsilon, \delta)$-differentially private and $\mathrm{Reg}_{1\text{-Select}_d}(\mathcal{M}) \leq \frac{\alpha}{n}$.*

*Proof.* **DP guarantee:** Fix neighboring datasets $\mathbf{y}$ and $\mathbf{y}'$ that are inputs to algorithm $\mathcal{M}$. According to the algorithm design for 1-Select$_d$, we stream $\mathbf{y}$ and $\mathbf{y}'$ to a DP-OPE algorithm $\mathcal{A}$. Since $\mathcal{A}$ is $(\varepsilon, \delta)$-DP, and $\mathcal{M}$ only post-processes the outputs received from $\mathcal{A}$, therefore $\mathcal{M}$ is $(\varepsilon, \delta)$-DP.

**Regret upper bound:** Fix a dataset $\mathbf{y}$. Note that if $\alpha \geq n$, the accuracy guarantee for $\mathcal{M}$ is vacuous. Now assume $\alpha < n$. Let $\gamma$ be the regret of $\mathcal{M}$, that is,

$$\gamma = \mathrm{Reg}_{1\text{-Select}_d}(\mathcal{M})$$
$$= \frac{1}{n}\left[\sum_{i=1}^{n} y_i(b) - \min_{x^\star \in [d]}\left(\sum_{i=1}^{n} y_i(x^\star)\right)\right],$$

where $b$ is the output of $\mathcal{M}$.

The main observation in the regret analysis is that if $\alpha$ is small, so is $\gamma$. Specifically, the regret of $\mathcal{M}$ is at most $\frac{\alpha}{n}$. Therefore, $\gamma \leq \frac{\alpha}{n}$. $\qquad\square$

**Lemma 11** (Jain et al. (2023), Lemma 4.2). *For all $d, n \in \mathbb{N}, \varepsilon \in (0, 1], \delta = o(1/n), \gamma \in \left[0, \frac{1}{20}\right]$, and $(\varepsilon, \delta)$-DP 1-Select$_d$ algorithms $\mathcal{M}: \left(\{0, 1\}^d\right)^n \to [d]$ with $\mathrm{Reg}_{1\text{-Select}_d}(\mathcal{M}) \leq \gamma$, we have $n = \Omega\left(\frac{\log d}{\varepsilon \gamma}\right)$.*

# E  Algorithms and Proofs for Fed-SVT

## E.1  Proof of Theorem 3

**Theorem 9** (Restatement of Theorem 3). *Let $\varepsilon \leq 1$ and $\delta \leq \varepsilon/d$. For any $(\varepsilon, \delta)$-DP federated OPE algorithm against oblivious adversaries in the realizable setting, the per-client regret is lower bounded by $\Omega\left(\frac{\log(d)}{m\varepsilon}\right)$.*

*Proof.* I introduce two prototype loss functions first: let $l^0(x) = 0$ for all $x \in [d]$ and for $j \in [d]$ let $l^j(x)$ be the function that has $l^j(x) = 0$ for $x = j$ and otherwise $l^j(x) = 1$. Then we define $d$ loss sequences $\mathcal{S}^j = (l^j_{1,1}, \ldots, l^j_{m,T})$, such that

$$l^j_{i,t} = \begin{cases} l^0 & \text{if} \quad t = 1, \ldots, T-k, \\ l^j & \text{else}; \end{cases}$$

where $k = \frac{\log d}{2m\varepsilon}$ and $j \in [d]$.

The oblivious adversary picks one of the $d$ sequences $\mathcal{S}^1, \ldots, \mathcal{S}^d$ uniformly at random. Assume towards a contradiction that $\mathbb{E}\left[\sum_{i=1}^m \sum_{t=1}^T l_{i,t}(x_{i,t}) - \sum_{i=1}^m \sum_{t=1}^T l_{i,t}(x^\star)\right] \leq \frac{\log(d)}{32\varepsilon}$. This implies that there exists $d/2$ sequences such that the expected regret satisfies $\mathbb{E}\left[\sum_{i=1}^m \sum_{t=1}^T l^j_{i,t}(x_{i,t}) - \sum_{i=1}^m \sum_{t=1}^T l^j_{i,t}(x^\star)\right] \leq \frac{\log(d)}{16\varepsilon}$. Assume without loss of generality these sequences are $\mathcal{S}^1, \ldots, \mathcal{S}^{d/2}$. Let $\mathcal{B}_j$ be the set of outputs that has low regret on $\mathcal{S}^j$, that is,

$$\mathcal{B}_j = \left\{ (x_{1,1}, \ldots, x_{m,T}) \in [d]^{mT} : \sum_{i=1}^m \sum_{t=1}^{T-k+1} \ell^j(x_{i,t}) \leq \frac{\log(d)}{8\varepsilon} \right\}.$$

Note that $\mathcal{B}_j \cap \mathcal{B}_{j'} = \emptyset$ since if $x_{1:m,1:T} \in B_j$ then among the $mk$ outputs $x_{1:m,T-k+1:T}$ at least $\frac{3mk}{4} = \frac{3\log(d)}{8\varepsilon}$ of them must be equal to $j$. Now Markov inequality implies that

$$\mathbb{P}\left(\mathcal{A}\left(\mathcal{S}^j\right) \in \mathcal{B}_j\right) \geq 1/2.$$

Moreover, group privacy gives

$$\mathbb{P}\left(\mathcal{A}\left(\mathcal{S}^j\right) \in \mathcal{B}_{j'}\right) \geq e^{-mk\varepsilon}\mathbb{P}\left(\mathcal{A}\left(\mathcal{S}^{j'}\right) \in \mathcal{B}_{j'}\right) - mk\delta$$
$$\geq \frac{1}{2\sqrt{d}} - \frac{\log d}{2\varepsilon}\delta$$
$$\geq \frac{1}{4\sqrt{d}}$$

where the last inequality is due to $\delta \leq \varepsilon/d$. Overall we get that

$$\frac{d/2 - 1}{4\sqrt{d}} \leq \mathbb{P}\left(\mathcal{A}\left(\mathcal{S}^j\right) \notin \mathcal{B}_j\right) \leq \frac{1}{2}$$

which is a contradiction for $d \geq 32$. $\qquad\square$

## E.2  Algorithm Design of Fed-SVT

Fed-SVT contains a client-side subroutine (Algorithm 7) and a server-side subroutine (Algorithm 8).

## E.3  Proof of Theorem 4

**Theorem 10** (Restatement of Theorem 4). *Let $l_{i,t} \in [0,1]^d$ be chosen by an oblivious adversary under near-realizability assumption. Set $0 < \rho < 1/2$, $\kappa = O(\log(d/\rho))$, $L = mL^\star + \frac{8\log(2T^2/(N^2\rho))}{\varepsilon} + 4/\eta$, and $\eta = \varepsilon/2\kappa$. Then the algorithm is $\varepsilon$-DP, the communication cost*

---

**Algorithm 7** Fed-SVT: Client $i$

---

1: **Input:** Number of Iterations $T$
2: **Initialize:** Set current expert $x_0 = \text{Unif}[d]$.
3: **for** $t = 1$ to $T$ **do**
4:     **if** $t == nN$ for some integer $n \geq 1$ **then**
5:         **Communicate to server:** $\sum_{t'=t-N}^{t-1} l_{i,t'}(x)$
6:         **Receive from server:** $x_t$
7:     **else**
8:         Set $x_t = x_{t-1}$
9:     **end if**
10:    Each client receives local loss $l_{i,t} : [d] \to [0,1]$ and pays cost $l_{i,t}(x_t)$
11: **end for**

---

---

**Algorithm 8** Fed-SVT: Central server

---

1: **Input:** Number of Iterations $T$, number of clients $m$, optimal loss $L^\star$, switching budget $\kappa$, sampling parameter $\eta > 0$, threshold parameter $L$, failure probability $\rho$, privacy parameters $\varepsilon$
2: **Initialize:** Set $k = 0$, $\tau = 0$ and $\hat{L} = L + \text{Lap}\left(\frac{4}{\varepsilon}\right)$
3: **while** not reaching the time horizon $T$ **do**
4:     **if** $t == nN$ for some integer $n \geq 1$ **then**
5:         **Receive from clients:** $\sum_{t'=t-N}^{t-1} l_{i,t'}(x)$
6:         **if** $k < \kappa$ **then**
7:             Server defines a new query $q_t = \sum_{i=1}^{m} \sum_{t'=\tau}^{t-1} l_{i,t'}(x_{t'})$
8:             Let $\gamma_t = \text{Lap}\left(\frac{8}{\varepsilon}\right)$
9:             **if** $q_t + \gamma_t \leq \hat{L}$ **then**
10:                **Communicate to clients:** $x_t = x_{t-1}$
11:             **else**
12:                Sample $x_t$ with scores $s_t(x) = \max\left(\sum_{i=1}^{m} \sum_{t'=1}^{t-1} l_{i,t'}(x_{t'}), mL^\star\right)$ for $x \in [d]$:
$$\mathbb{P}(x_t = x) \propto e^{-\eta s_t(x)/2}$$
13:                **Communicate to clients:** $x_t$
14:                Set $k = k + 1$, $\tau = t$ and $\hat{L} = L + \text{Lap}\left(\frac{4}{\varepsilon}\right)$
15:             **end if**
16:         **else**
17:             Server broadcasts $x_t = x_{t-1}$ to all the clients
18:         **end if**
19:     **end if**
20: **end while**

---

scales in $O\left(mdT/N\right)$, and with probability at least $1 - O(\rho)$, the pre-client regret is upper bounded by $O\left(\frac{\log^2(d) + \log\left(\frac{T^2}{N^2\rho}\right)\log\left(\frac{d}{\rho}\right)}{m\varepsilon} + (N + L^\star)\log\left(\frac{d}{\rho}\right)\right)$.

*Moreover, setting $\eta = \varepsilon/\sqrt{\kappa \log(1/\delta)}$, we have the algorithm is $(\varepsilon, \delta)$-DP, the communication cost scales in $O\left(mdT/N\right)$, and with probability at least $1 - O(\rho)$, the pre-client regret is upper bounded by* $O\left(\frac{\log^{\frac{3}{2}}(d)\sqrt{\log(\frac{1}{\delta})} + \log\left(\frac{T^2}{N^2\rho}\right)\log\left(\frac{d}{\rho}\right)}{m\varepsilon} + (N + L^\star)\log\left(\frac{d}{\rho}\right)\right)$.

*Proof.* **DP guarantee:** There are $\kappa$ applications of exponential mechanism with privacy parameter $\eta$. Moreover, sparse vector technique is applied over each sample once, hence the $\kappa$ applications of sparse-vector are $\varepsilon/2$-DP. Overall, the algorithm is $(\varepsilon/2 + \kappa\eta)$-DP and $(\varepsilon/2 + \sqrt{2\kappa \log(1/\delta)}\eta + \kappa\eta(e^\eta - 1), \delta)$-DP (Lemma 14). Setting $\eta = \varepsilon/2\kappa$ results in $\varepsilon$-DP and $\eta = \varepsilon/\sqrt{\kappa \log(1/\delta)}$ results in $(\varepsilon, \delta)$-DP.

**Communication cost:** The number of communication between the central server and clients scales in $O(mT/N)$. Moreover, within each communication, the number of scalars exchanged scales in $O(d)$. Therefore the communication cost is $O(mdT/N)$.

**Regret upper bound:** We define a potential at phase $n \in [T/N]$ :

$$\phi_n = \sum_{x \in [d]} e^{-\eta L_n(x)/2}$$

where $L_n(x) = \max\left(\sum_{i=1}^m \sum_{t'=1}^{nN-1} l_{i,t'}(x), mL^\star\right)$. Note that $\phi_0 = de^{-\eta m L^\star/2}$ and $\phi_n \geq e^{-\eta m L^\star/2}$ for all $n \in [T/N]$ since there is $x \in [d]$ such that $\sum_{i=1}^m \sum_{t=1}^T l_{i,t}(x) \leq mL^\star$. We split the iterates to $s = \lceil \log d \rceil$ rounds $n_0 N, n_1 N, \ldots, n_s N$ where $n_p$ is the largest $n \in [T/N]$ such that $\phi_{n_p} \geq \phi_0/2^p$. Let $Z_p$ be the number of switches in $[n_p N, (n_{p+1}-1)N]$ (number of times the exponential mechanism is used to pick $x_t$). Let $Z = \sum_{p=0}^{s-1} Z_p$ be the total number of switches. Note that $Z \leq 3s + \sum_{p=0}^{s-1} \max(Z_p - 3, 0)$ and Lemma 12 implies $\max(Z_p - 3, 0)$ is upper bounded by a geometric random variable with success probability $1/3$. Therefore, using concentration of geometric random variables (Lemma 13), we get that

$$P(Z \geq 3s + 24 \log(1/\rho)) \leq \rho.$$

Since $K \geq 3s + 24 \log(1/\rho)$, the algorithm does not reach the switching budget with probability $1 - O(\rho)$. So the total number of switching scales as $O(\log(d/\rho))$. Now we analyze the regret. Define $T_1 N, \ldots T_C N$ as the switching time steps with $T_C N = T$, where $C = O(\log(d/\rho))$. Lemma 16 implies that with probability at least $1 - \rho$,

$$
\begin{aligned}
\sum_{t=1}^T \sum_{i=1}^m l_{i,t}(x_{i,t}) &= \sum_{c=1}^C \sum_{t=T_{c-1}N+1}^{T_c N} \sum_{i=1}^m l_{i,t}(x_{i,t}) \\
&= \sum_{c=1}^C \left( \sum_{t=T_{c-1}N+1}^{T_c N - N} \sum_{i=1}^m l_{i,t}(x_{i,t}) + \sum_{t=T_c N - N + 1}^{T_c N} \sum_{i=1}^m l_{i,T_n}(x_t) \right) \\
&\leq \sum_{c=1}^C \left( L + \frac{8 \log(2T^2/(N^2 \rho))}{\epsilon} + mN \right) \\
&= \sum_{c=1}^C \left( mL^\star + \frac{16 \log(2T^2/(N^2 \rho))}{\epsilon} + 4/\eta + mN \right) \\
&= O\left( mL^\star \log(d/\rho) + \frac{\log(T^2/(N^2 \rho)) \log(d/\rho)}{\varepsilon} + \frac{4 \log(d/\rho)}{\eta} + mN \log(d/\rho) \right).
\end{aligned}
$$

$$(8)$$

**Case 1:** Setting $\eta = \varepsilon/2\kappa$ in Equation (8), we have

$$\sum_{i=1}^m \sum_{t=1}^T l_{i,t}(x_{i,t}) \leq O\left( mL^\star \log d + \frac{\log^2 d + \log(T^2/(N^2\rho)) \log(d/\rho)}{\varepsilon} + mN \log(d/\rho) \right).$$

**Case 2:** Setting $\eta = \varepsilon/\sqrt{\kappa \log(1/\delta)}$ in Equation (8), we have

$$\sum_{i=1}^m \sum_{t=1}^T l_{i,t}(x_{i,t}) \leq O\left( mL^\star \log d + \frac{\log^{3/2} d \sqrt{\log(1/\delta)} + \log(T^2/(N^2\rho)) \log(d/\rho)}{\varepsilon} + mN \log(d/\rho) \right).$$

**Lemma 12.** *Fix $0 \leq p \leq s - 1$. Then for any $1 \leq k \leq T/N$, it holds that*

$$P(Z_p = k + 3) \leq (2/3)^{k+2} < (2/3)^{k-1}(1/3).$$

*Proof.* Let $n_p N \le nN \le n_{p+1}N$ be a time-step when a switch happens (exponential mechanism is used to pick $x_t$). Note that $\phi_{n_{p+1}} \ge \phi_n/2$. We prove that the probability that $x_t$ is switched between $nN$ and $n_{p+1}N$ is at most $2/3$. To this end, note that if $x_t$ is switched before $n_{p+1}N$ then $\sum_{i=1}^{m}\sum_{t'=nN}^{n_{p+1}N-1} l_{i,t'}(x_{t'}) \ge L - \frac{8\log(2T^2/(N^2\rho))}{\varepsilon}$, therefore $L_{n_{p+1}}(x) - L_n(x) \ge L - \frac{8\log(2T^2/(N^2\rho))}{\varepsilon} \ge 4/\eta$. Thus we have that

$$
\begin{aligned}
P\left(x_t \text{ is switched before } n_{p+1}N\right) &\le \sum_{x\in[d]} P\left(x_t = x\right) \mathbf{1}\left\{L_{n_{p+1}}(x) - L_n(x) \ge 4/\eta\right\} \\
&= \sum_{x\in[d]} \frac{e^{-\eta L_n(x)/2}}{\phi_n} \cdot \mathbf{1}\left\{L_{n_{p+1}}(x) - L_n(x) \ge 4/\eta\right\} \\
&\le \sum_{x\in[d]} \frac{e^{-\eta L_n(x)/2}}{\phi_n} \cdot \frac{1 - e^{-\eta\left(L_{n_{p+1}}(x) - L_n(x)\right)/2}}{1 - e^{-2}} \\
&\le 4/3\left(1 - \phi_{n_{p+1}}/\phi_n\right) \\
&\le 2/3.
\end{aligned}
$$

where the second inequality follows the fact that $\mathbf{1}\{a \ge b\} \le \frac{1-e^{-\eta a}}{1-e^{-\eta b}}$ for $a, b, \eta \ge 0$, and the last inequality since $\phi_{n_{p+1}}/\phi_n \ge 1/2$. This argument shows that after the first switch inside the range $[n_p N, n_{p+1}N - 1]$, each additional switch happens with probability at most $2/3$. So we have

$$
P\left(Z_p = k + 3\right) \le (2/3)^{k+2} < (2/3)^{k-1}(1/3).
$$

$\square$

**Lemma 13** (Asi et al. (2023), Lemma A.2). *Let $W_1, \ldots, W_n$ be i.i.d. geometric random variables with success probability $p$. Let $W = \sum_{i=1}^{n} W_i$. Then for any $k \ge n$*

$$
\mathbb{P}(W > 2k/p) \le \exp(-k/4).
$$

$\square$

## F  Experimental Supplementary

The simulations were conducted on a system with a 2.3 GHz Dual-Core Intel Core i5, Intel Iris Plus Graphics 640 with 1536 MB, and 16 GB of 2133 MHz LPDDR3 RAM. Approximately 10 minutes are required to reproduce the experiments.

We present our numerical results with different seeds in Figure 4 and Figure 5.

## G  Experiments on MovieLens-1M

We use the MovieLens-1M dataset (Harper and Konstan, 2015) to evaluate the performances of Fed-SVT, comparing it with the single-player model Sparse-Vector (Asi et al., 2023). We first compute the rating matrix of 6040 users to 18 movie genres (experts) $R = [r_{u,g}] \in \mathbb{R}^{6040\times 18}$, and then calculate $L = [\max(0, r_{u,g^\star} - r_{u,g})] \in \mathbb{R}^{6040\times 18}$ where $g^\star = \arg\max_g \left(\frac{1}{6040}\sum_{u=1}^{6040} r_{u,g}\right)$. We generate the sequence of loss functions $\{l_u\}_{u\in[6040]}$ where $l_u = L_{u,:}$. In our experiments, we set $m = 10$, $T = 604$, $\varepsilon = 10$, $\delta = 0$ and run 10 trials. In Fed-SVT, we experiment with communication intervals $N = 1, 30, 50$, where communication cost scales in $O(mdT/N)$. The per-client cumulative regret as a function of $T$ is plotted in Figure 6. Our results show that Fed-SVT significantly outperforms Sparse-Vector with low communication costs (notably in the $N = 50$ case). These results demonstrate the effectiveness of our algorithm in real-world applications.

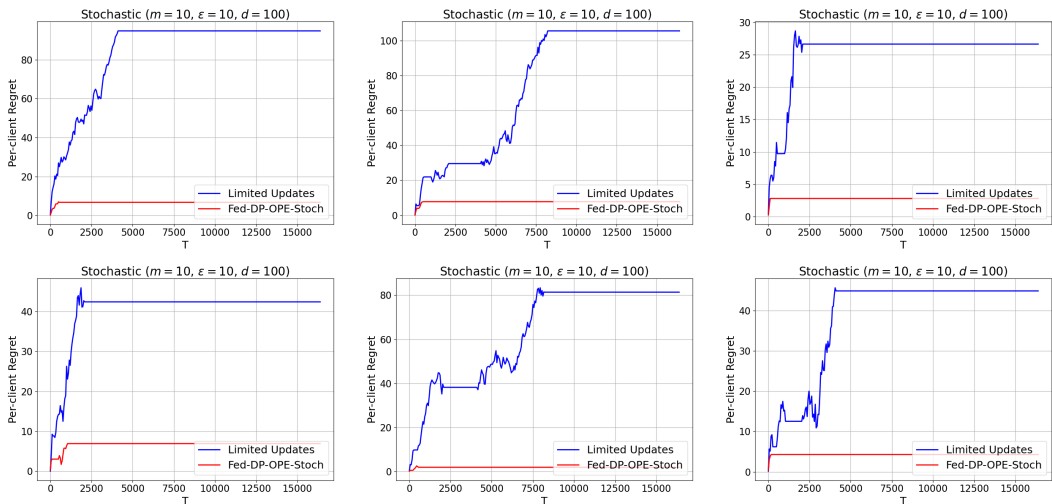

Figure 4: Comparison between Fed-DP-OPE-Stoch and Limited Updates with different random seeds.

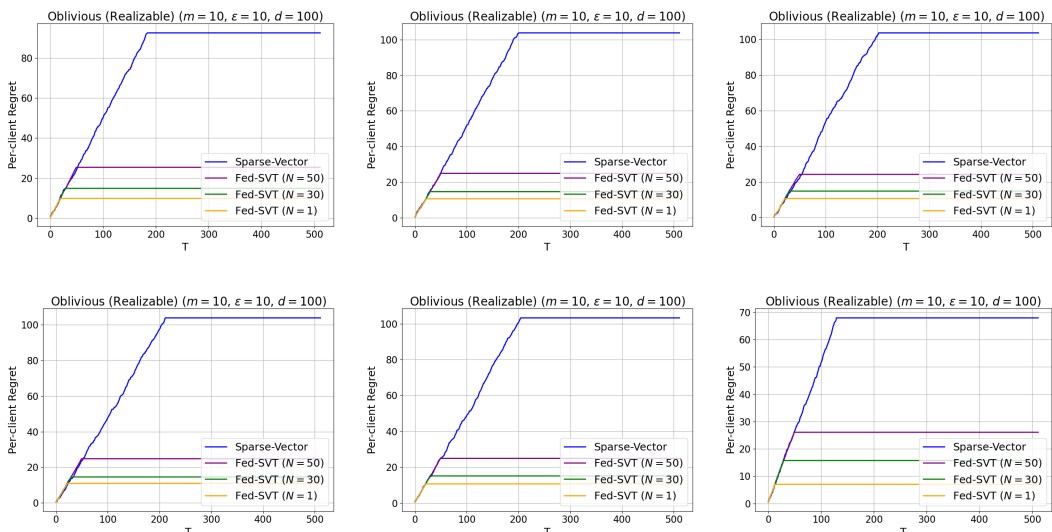

Figure 5: Comparison between Fed-SVT and Sparse-Vector with different random seeds.

# H  Background on Differential Privacy

## H.1  Advanced Composition

**Lemma 14** (Advanced composition, Dwork et al. (2014)). *If $\mathcal{A}_1, \ldots, \mathcal{A}_k$ are randomized algorithms that each is $(\varepsilon, \delta)$-DP, then their composition $(\mathcal{A}_1(\mathcal{S}), \ldots, \mathcal{A}_k(\mathcal{S}))$ is $(\sqrt{2k \log(1/\delta')}\varepsilon + k\varepsilon(e^\varepsilon - 1), \delta' + k\delta)$-DP.*

## H.2  Report Noisy Max

The "Report Noisy Max" mechanism can be used to privately identify the counting query among $m$ queries with the highest value. This mechanism achieves this by adding Laplace noise independently generated from $\mathrm{Lap}(\Delta/\varepsilon)$ to each count and subsequently determining the index corresponding to the largest noisy count (we ignore the possibility of a tie), where $\Delta$ is the sensitivity of the queries. Report noisy max gives us the following guarantee.

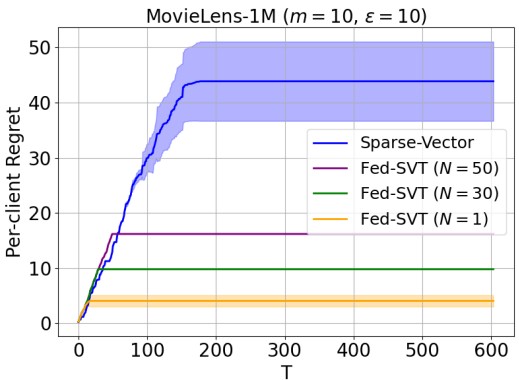

Figure 6: Regret performance with MovieLens dataset. Shaded area indicates the standard deviation.

**Lemma 15** (Dwork et al. (2014), Claim 3.9). *The Report Noisy Max algorithm is $\varepsilon$-differentially private.*

### H.3 Sparse Vector Technique

We recall the sparse-vector technique here. Given an input $\mathcal{S} = (z_1, \ldots, z_n) \in \mathcal{Z}^n$, the algorithm takes a stream of queries $q_1, q_2, \ldots, q_T$ in an online manner. We assume that each $q_i$ is 1-sensitive, i.e., $|q_i(\mathcal{S}) - q_i(\mathcal{S}')| \leq 1$ for neighboring datasets $\mathcal{S}, \mathcal{S}' \in \mathcal{Z}^n$ that differ in a single element. We have the following guarantee.

**Lemma 16** (Dwork et al. (2014), Theorem 3.24). *Let $\mathcal{S} = (z_1, \ldots, z_n) \in \mathcal{Z}^n$. For a threshold $L$ and $\rho > 0$, there is an $\varepsilon$-DP algorithm (AboveThreshold) that halts at time $k \in [T+1]$ such that for $\alpha = \frac{8(\log T + \log(2/\rho))}{\varepsilon}$ with probability at least $1 - \rho$, we have $q_i(\mathcal{S}) \leq L + \alpha$ for all $t < k$, and $q_k(\mathcal{S}) \geq L - \alpha$ or $k = T + 1$.*

## I Broader Impacts

This work improves online learning and decision-making through collaboration among multiple users without exposing personal information, which helps balance the benefits of big data with the need to protect individual privacy, promoting ethical data usage and fostering societal trust in an increasingly data-driven world.

