# OpenReview forum: "Federated Online Prediction from Experts with Differential Privacy: Separations and Regret Speed-ups"
_NeurIPS.cc/2024/Conference — NeurIPS 2024 poster_

### Official Review · Reviewer_hhm9 · 2024-07-12

**Soundness:** 4
**Presentation:** 3
**Contribution:** 3
**Rating:** 7
**Confidence:** 3

**Summary:**

The paper addresses the problem of online prediction from experts (OPE) under differential privacy (DP) constraints in a federated setting. OPE operates in a set of rounds, and consists on choosing at each round the expert that minimizes the regret over observations of data. The selection of the expert at each round is based on previous adversarially chosen observations of data. Differentially private OPE (DP-OPE) ensures that the selection of experts is not significantly different (up to $\epsilon$ and $\delta$ parameters) if one observation of the data changes.

The current protocol studies private OPE in the federated setting (Fed-DP-OPE), where the expert selection is done by a set of clients that collaborate via a server. Under local DP (i.e. all messages between server and clients are part of the output of the protocol) and $m$ clients, the current work provides fundamental bounds of over the impact of the federated collaboration on the regret. First, it proposes a protocol that achieves an order-wise $\sqrt{m}$ multiplicative regret reduction with respect to DP-OPE when the adversarial observations are chosen randomly. Second, it shows that the communication cost is logarithmic in the number of expert selection rounds. Third, it shows that under an oblivious adversary that chooses the the observations arbitrarily in advance, it is not possible to obtain an order-wise improvement regret in Fed-DP-OPE over DP-OPE. Finally, it shows that improvement is possible in the federated setting for a relaxation of the oblivious adversary.

**Strengths:**

The paper is in general clearly written.

The contributions provide a significant impact, showing that federated collaboration can improve accuracy under privacy constraints without incurring in large communication cost (i.e. the cost is logarithmic in $T$ for the stochastic adversary setting). The $\sqrt{m}$ factor speedup is significant. The relaxation of the oblivious adversary seems realistic, therefore improvements in this setting seem also important.

The paper also proposes novel techniques. Although the natural adaptation to federated setting and the use tree based private aggregation have been widely studied in the past, the paper also proposes novel ways to reduce (i) communication in the stochastic adversary setting, (ii)  regret and communication in the (relaxed) oblivious adversary setting and (iii) a novel proof technique for the lower bounds on the classical oblivious case.

The work is of good quality, appropriately backing it's claims with proofs and empirically illustrating the impact of the results with reproducible experiments.

**Weaknesses:**

There are certain non-major weaknesses in motivating the problem and clarity at preliminaries:

1- The implications of accurate OPE could be better motivated in the introduction.

2- The notion of DP in the online setting could be better clarified. First, a link between the chosen sequence of loss functions and possibly sensitive data is missing. Second, the definition specifies "DP against an adversary". The same terminology for DP adversary and OPE adversary in charge of selecting observations over the sensitive dataset is confusing.

**Questions:**

Could you please clarify what is the link between sensitive data and the notion of neighboring input used in the paper?

For instance, it would seem reasonable that the sequence $\mathcal{S}$ of loss functions chosen by the adversary depend on a set of sensitive data points. Then, it could be possible that more than one loss function changes if a single entry of this dataset changes. However, your setting is a bit different and I am wondering where could it be applicable.

**Limitations:**

The limitations of the work have been appropriately addressed by the authors.

---

> ### Author Rebuttal · Authors · 2024-08-07
>
> We thank the reviewer for the careful reading and thoughtful comments. We address the reviewer's questions in the following and will revise our paper accordingly. We hope the responses below address the reviewer's concerns.
>
>
> **Q1:** There are certain non-major weaknesses in motivating the problem and clarity at preliminaries: The implications of accurate OPE could be better motivated in the introduction.
>
> **A1:** Thank you for your helpful comment. We will clarify the implications of accurate OPE in the introduction as follows:
>
> OPE is an important problem in machine learning with various applications. For example, in personalized healthcare, a patient's wearable device (player) collects and processes health data to provide personalized health treatment (experts). OPE helps select the most effective treatments based on historical data, improving patients' outcomes.
>
> We will include this to better highlight the implications of OPE in our revised introduction.
>
> **Q2:** The notion of DP in the online setting could be better clarified. First, a link between the chosen sequence of loss functions and possibly sensitive data is missing. Second, the definition specifies "DP against an adversary". The same terminology for DP adversary and OPE adversary in charge of selecting observations over the sensitive dataset is confusing.
>
> **A2:** Thank you for your insightful comments. We will clarify the notion of DP in the online setting in the revision.
>
> 1. Link Between Loss Functions and Sensitive Data: The sequence of loss functions $\mathcal{S}=\left(l_{1,1}, \ldots, l_{m,T}\right)$ represents the sensitive dataset since each loss function $l_{i,t}$ reflects the performance of selecting different experts based on sensitive information about client $i$ at time step $t$.
>
> 2. Terminology Confusion: To clarify, the OPE adversary refers to the entity selecting the sequence of loss functions to challenge the online algorithm, while the DP adversary represents a potential attacker trying to infer sensitive information from the algorithm's outputs. We will revise the preliminary section to distinguish these roles clearly, referring to the OPE adversary as the "game adversary" and the DP adversary as the "privacy adversary".
>
> **Q3:** Could you please clarify what is the link between sensitive data and the notion of neighboring input used in the paper? For instance, it would seem reasonable that the sequence of loss functions chosen by the adversary depend on a set of sensitive data points. Then, it could be possible that more than one loss function changes if a single entry of this dataset changes. However, your setting is a bit different and I am wondering where could it be applicable.
>
> **A3:** Thank you for the question. In our setting, the sequence of loss functions $\mathcal{S}=\left(l_{1,1}, \ldots, l_{m,T}\right)$ is considered the sensitive dataset and we define neighboring datasets as sequences that differ by a single loss function. Our setting is applicable when protecting individual loss functions is sufficient. For example, in a personalized recommendation system, each user's interaction with content (e.g. articles) generates a loss function. Our setting ensures that changing a single interaction does not significantly influence the system's recommendations, protecting individual user interactions. This is suitable for applications like personalized content recommendations, where protecting individual interactions is essential while still allowing for effective personalization.
>
> We understand that the reviewer's example describes a stronger DP setting, where one sensitive data point may affect multiple loss functions, and all of those loss functions need to be protected simultaneously. In some sense, we can view one data point as a user, and each user may contribute multiple loss functions. Thus, to protect individual users, it is necessary to protect all loss functions that each user contributes. This essentially gives rise to **user-level DP** [1]. We note that user-level DP in general is much more challenging to handle [2-6] compared with instance-level DP discussed in this work, and we leave it as one of the important future directions to explore.
>
> [1] Levy D, Sun Z, Amin K, et al. Learning with user-level privacy[J]. Advances in Neural Information Processing Systems, 2021, 34: 12466-12479.
>
> [2] Liu Y, Suresh A T, Yu F X X, et al. Learning discrete distributions: user vs item-level privacy[J]. Advances in Neural Information Processing Systems, 2020, 33: 20965-20976.
>
> [3] Acharya J, Liu Y, Sun Z. Discrete distribution estimation under user-level local differential privacy[C]//International Conference on Artificial Intelligence and Statistics. PMLR, 2023: 8561-8585.
>
> [4] Cummings R, Feldman V, McMillan A, et al. Mean estimation with user-level privacy under data heterogeneity[J]. Advances in Neural Information Processing Systems, 2022, 35: 29139-29151.
>
> [5] Ghazi B, Kumar R, Manurangsi P. User-level differentially private learning via correlated sampling[J]. Advances in Neural Information Processing Systems, 2021, 34: 20172-20184.
>
> [6] Huang R, Zhang H, Melis L, et al. Federated linear contextual bandits with user-level differential privacy[C]//International Conference on Machine Learning. PMLR, 2023: 14060-14095.
>
> -----
>
> We thank the reviewer again for the helpful comments and suggestions for our work. We are more than happy to address any further questions that you may have.

---

> ### Author Response · Authors · 2024-08-12
>
> Dear Reviewer hhm9,
>
> We've taken your initial feedback into careful consideration in our response. Could you please check whether our responses have properly addressed your concerns? We are more than happy to answer your further questions.
>
> Thank you for your time and effort in reviewing our work!
>
> Best Regards,
>
> Authors

---

> > ### Comment · Reviewer_hhm9 · 2024-08-13
> >
> > Dear authors,
> >
> > Thank you for successfully addressing my (non-major) concerns in your rebuttal.
> >
> > I have no further questions and my score is likely to remain unchanged.

---

> > > ### Author Response · Authors · 2024-08-13
> > >
> > > Thank you so much for your positive feedback! Your insightful comments have helped us make our work better. We really appreciate it.

---

### Official Review · Reviewer_RFE1 · 2024-07-13

**Soundness:** 3
**Presentation:** 3
**Contribution:** 3
**Rating:** 5
**Confidence:** 3

**Summary:**

The paper studies the problem of online federated expert selection. In order to make the proposed algorithm robust against adversaries, the paper proposes algorithms with differential privacy guarantees. Both stochastic and oblivious adversaries are investigated by the paper. The paper provides theoretical guarantees for the proposed algorithms. The paper provides experimental results to verify theoretical findings.

**Strengths:**

1. The paper proposes differentially private online federated learning algorithm which is an important and unexplored research area.
2. The paper provides regret analysis with differential privacy guarantees for the proposed algorithms. Although I am not expert at differential privacy, theoretical results looks convincing to me.
3. The paper is well-written and clear.

**Weaknesses:**

1. The paper can benefit from extending its experimental study. It would be great if authors can add more datasets and also do more ablation studies to strengthen the contribution of the paper.
2. The paper can add more explanations about applicability of the study. Usually federated learning is used to train a model. However, reading the paper, the practical applications of the paper is not obvious.

**Questions:**

1. I do not understand the statement in the paper that "*In the federated setting, it is infeasible for the central server to have full access to past loss functions due to the high communication cost.*" I think in these cases the server can collect losses over time and store them in such a way that clients can afford communication cost. I believe in these cases probably storage can be more important bottleneck than communication cost. Furthermore, I do not think sending a loss would be bottleneck for the system in both communication and storage aspects even if the number of clients is large. Can you explain more about this?
2. I suggest extending the literature review of the paper by including the work "Personalized online federated learning with multiple kernels. *Advances in Neural Information Processing Systems (NeurIPS)*, 35:33316–33329, 2022.". This work studies online model selection for communication and prediction efficiency in online federated learning.

**Limitations:**

The paper discusses the limitations adequately. For example, the paper clearly presents the assumption in section 3 that the loss functions are assumed to be convex and smooth with respect to $||\cdot||_1$.

---

> ### Author Rebuttal · Authors · 2024-08-06
>
> We thank the reviewer for the careful reading and thoughtful comments. We address the reviewer's questions in the following. We hope the responses below address the reviewer's concerns.
>
> **Q1:** The paper can benefit from extending its experimental study.
>
> **A1:** Thanks for the helpful suggestion. We have performed experiments on a real-world dataset. Please refer to our Author Rebuttal.
>
> **Q2:** The paper can add more explanations about applicability of the study.
>
> **A2:** Thanks for the insightful suggestion. In this work, we focus on differentially private federated online prediction, where the objective is to minimize the average regret on $m$ clients working in parallel while maintaining the privacy of users at each client. Differentially private federated online prediction has many important real-world applications. We provide three examples below, and will add the discussions in our revision.
>
> **Personalized Healthcare**: Consider a federated online prediction setting where patients' wearable devices collect and process health data locally, and the central server aggregates privacy-preserving updates from devices to provide health recommendations or alerts. DP federated online prediction can speed up the learning process and improve the prediction accuracy without exposing individual's health data, thus ensuring patient privacy.
>
> **Financial Fraud Detection**: DP federated online prediction can also enhance fraud detection systems across banking and financial services. Each client device (e.g. PC) locally analyzes transaction patterns and flags potential fraud without revealing sensitive transaction details to the central server. The server's role is to collect privacy-preserving updates from these clients to improve the global fraud detection model. This method ensures that the financial company can dynamically adapt to new fraudulent tactics, improving detection rates while safeguarding customers' financial privacy.
>
> **Personalized Recommender Systems**: Each client (e.g. smartphone) can personalize content recommendations by analyzing user interactions and preferences locally. The central server (e.g. company) aggregates privacy-preserving updates from all clients to refine the recommendation model. Thus, DP federated online prediction improves the whole recommender system performance while maintaining each client's privacy.
>
> **Q3:** Can you explain more about the statement that "In the federated setting, it is infeasible for the central server to have full access to past loss functions due to the high communication cost"?
>
> **A3:** Thank you for your insightful comments. In federated learning, training data is distributed across an incredibly large number of devices, each potentially having limited communication bandwidth to the central server [1]. Consider the hospitals within an area as the set of clients, each aiming to make accurate predictions (e.g. diagnostic) for its own patients. Then, the loss function for each client depends on the healthcare history of all patients in the hospital. Obviously, transmitting patients' healthcare histories from all hospitals to a central server frequently requires a lot of communication bandwidth and is often time-consuming. One major motivation for federated learning is to avoid transmitting raw data (e.g., loss functions) to the central server, while maintaining learning performances comparable with the centralized setting [2-5].
>
> Moreover, keeping raw data (e.g., loss functions) at local clients in federated learning is also beneficial for **privacy protection**. Intuitively, to protect the loss function transmitted every round, we need to apply DP mechanism to privatize each loss function (e.g. Laplacian noise $\text{Lap}(\frac{1}{\varepsilon})$), as exposing sensitive patient data is a concern. Such a noise level will compromise the data accuracy at the server, and leads to significant learning performance degradation. In contrast, under an efficient communication protocol of federated learning, the noise level added during each communication round can be significantly reduced, since we don't have to protect each loss function but certain information extracted from multiple loss functions. For example, in Fed-DP-OPE-Stoch, each client adds Laplacian noise with standard deviation $O(\frac{1}{2^{p-1}\varepsilon})$ to the estimated gradients before transmission in phase $p\in [\log T]$. Such reduced noise level has minimum impact on the aggregated gradient estimate at the server, and results in near-optimal privacy-utility trade-off.
>
> In summary, due to both communication cost and privacy considerations, designing communication-efficient learning algorithms is of critical importance in the federated setting. In this work, we follow the standard federated learning principle to keep raw data (loss functions) at the clients, and design communication-efficient federated online prediction algorithms. We thank the reviewer again for the comment and will revise the statement in the paper to avoid possible confusion.
>
> **Q4:** I suggest extending the literature review of the paper by including the work "Personalized online federated learning with multiple kernels".
>
> **A4:** Thank you for the suggestion. We will include the recommended work and provide a literature review on (federated) online model selection in Appendix A as follows.
>
> **(Federated) Online Model Selection:** Online model selection when models are kernels has been extensively studied [6-9]. In the federated setting, [10,11] have explored scenarios where each client learns a kernel-based model, utilizing the specific characteristics of kernel functions. Moreover, [12] proposes an online federated model selection framework, where clients interact with a server with sufficient memory. Additionally, research on online learning with feedback graphs, a generalization of sequential decision-making with bandit or full information feedback, has been explored in [13-19].

---

> > ### Comment · Reviewer_RFE1 · 2024-08-13
> >
> > Thank you for responding to my comments. I read the rebuttal and it addressed part of my concerns. However, I believe the experimental study of the paper can still be extended and improved. I would maintain my initial rating which is in overall in favor of accepting the paper.

---

> > > ### Author Response · Authors · 2024-08-13
> > >
> > > Thank you for taking the time to review our rebuttal. We’re glad to hear that we addressed some of your concerns and that your overall assessment is in favor of accepting the paper.
> > >
> > > We understand your point regarding the experimental study. Due to time constraints, we were only able to conduct experiments on MovieLens dataset during rebuttal. We will certainly include additional experimental results in the revision.
> > >
> > > Thank you again for your feedback, and we are happy to address any further questions you might have.

---

> ### Author Response · Authors · 2024-08-06
> **References**
>
> [1] Li X, Huang K, Yang W, et al. On the convergence of fedavg on non-iid data[J]. arXiv preprint arXiv:1907.02189, 2019.
>
> [2] Shahid O, Pouriyeh S, Parizi R M, et al. Communication efficiency in federated learning: Achievements and challenges[J]. arXiv preprint arXiv:2107.10996, 2021.
>
> [3] McMahan B, Moore E, Ramage D, et al. Communication-efficient learning of deep networks from decentralized data[C]//Artificial intelligence and statistics. PMLR, 2017: 1273-1282.
>
> [4] Smith V, Chiang C K, Sanjabi M, et al. Federated multi-task learning[J]. Advances in neural information processing systems, 2017, 30.
>
> [5] Sattler F, Wiedemann S, Müller K R, et al. Robust and communication-efficient federated learning from non-iid data[J]. IEEE transactions on neural networks and learning systems, 2019, 31(9): 3400-3413.
>
> [6] Yang T, Mahdavi M, Jin R, et al. Online kernel selection: Algorithms and evaluations[C]//Proceedings of the AAAI Conference on Artificial Intelligence. 2012, 26(1): 1197-1203.
>
> [7] Zhang X, Liao S, Xu J, et al. Regret bounds for online kernel selection in continuous kernel space[C]//Proceedings of the AAAI Conference on Artificial Intelligence. 2021, 35(12): 10931-10938.
>
> [8] Ghari P M, Shen Y. Graph-aided online multi-kernel learning[J]. Journal of Machine Learning Research, 2023, 24(21): 1-44.
>
> [9] Ghari P M, Shen Y. Online multi-kernel learning with graph-structured feedback[C]//International Conference on Machine Learning. PMLR, 2020: 3474-3483.
>
> [10] Hong S, Chae J. Communication-efficient randomized algorithm for multi-kernel online federated learning[J]. IEEE transactions on pattern analysis and machine intelligence, 2021, 44(12): 9872-9886.
>
> [11] M Ghari P, Shen Y. Personalized online federated learning with multiple kernels[J]. Advances in Neural Information Processing Systems, 2022, 35: 33316-33329.
>
> [12] Ghari P M, Shen Y. Budgeted Online Model Selection and Fine-Tuning via Federated Learning[J]. arXiv preprint arXiv:2401.10478, 2024.
>
> [13] Mannor S, Shamir O. From bandits to experts: On the value of side-observations[J]. Advances in Neural Information Processing Systems, 2011, 24.
>
> [14] Cohen A, Hazan T, Koren T. Online learning with feedback graphs without the graphs[C]//International Conference on Machine Learning. PMLR, 2016: 811-819.
>
> [15] Alon N, Cesa-Bianchi N, Dekel O, et al. Online learning with feedback graphs: Beyond bandits[C]//Conference on Learning Theory. PMLR, 2015: 23-35.
>
> [16] Cortes C, DeSalvo G, Gentile C, et al. Online learning with dependent stochastic feedback graphs[C]//International Conference on Machine Learning. PMLR, 2020: 2154-2163.
>
> [17] Ghari P M, Shen Y. Online learning with uncertain feedback graphs[J]. IEEE Transactions on Neural Networks and Learning Systems, 2023.
>
> [18] Esposito E, Fusco F, van der Hoeven D, et al. Learning on the edge: Online learning with stochastic feedback graphs[J]. Advances in Neural Information Processing Systems, 2022, 35: 34776-34788.
>
> [19] Ghari P M, Shen Y. Online learning with probabilistic feedback[C]//ICASSP 2022-2022 IEEE International Conference on Acoustics, Speech and Signal Processing (ICASSP). IEEE, 2022: 4183-4187.

---

> ### Author Response · Authors · 2024-08-06
> **Thank You for Your Helpful Feedback**
>
> We thank the reviewer again for the helpful comments and suggestions for our work. If our response resolves your concerns to a satisfactory level, we kindly ask the reviewer to consider raising the rating of our work. Certainly, we are more than happy to address any further questions that you may have.

---

> ### Author Response · Authors · 2024-08-12
>
> Dear Reviewer RFE1,
>
> We've taken your initial feedback into careful consideration in our response. Could you please check whether our responses have properly addressed your concerns? If so, could you please kindly consider increasing your initial score accordingly? Certainly, we are more than happy to answer your further questions.
>
> Thank you for your time and effort in reviewing our work!
>
> Best Regards,
>
> Authors

---

### Official Review · Reviewer_NyWa · 2024-07-27

**Soundness:** 3
**Presentation:** 3
**Contribution:** 3
**Rating:** 6
**Confidence:** 3

**Summary:**

This paper studies differentially private federated online prediction from experts against stochastic and oblivious adversaries. The goal is to minimize average regret across clients over time with privacy guarantees. For stochastic adversaries, the proposed Fed-DP-OPE-Stoch algorithm achieves regret improvement over single-player counterparts with logarithmic communication costs. For oblivious adversaries with a low-loss expert, the new Fed-SVT algorithm demonstrates an m-fold regret speed-up under pure and approximate differential privacy, nearly optimal up to logarithmic factors according to established lower bounds. Simulation experiments have been done.

**Strengths:**

1, The paper is well written, presented: The background, related previous work, problem formulation and key concepts (e.g., federated online prediction, differential privacy, stochastic/oblivious adversaries) are clearly explained. The algorithms are described in detail.

2,  The theoretical analysis appears rigorous, with the authors establishing regret bounds for their proposed algorithms and deriving  lower bounds for the oblivious adversary case. The Fed-SVT algorithm achieves near-optimal regret performance (up to logarithmic factors) in the special case of oblivious adversaries with a low-loss expert, demonstrating the quality of the proposed solution.

**Weaknesses:**

1, Lack of Real-World Evaluation: The lack of experiments on real-world dataset scenarios is a significant limitation. The authors should address how their proposed algorithms would perform and be applicable to practical problems in domains like recommender systems or healthcare. Without evaluations on real-world datasets, it is challenging to judge the practical utility and potential challenges of their methods.

2, Novelty Concerns:  After given a tree-based method is used for private aggregation of the gradients of loss functions (Asi et al., 2021b), what's the challenge for the theoretical analysis of proposed algorithms given Asi et al., 2022b and Asi et al., 2023?

**Questions:**

1, Please add at least one experiment with real-world dataset. So reader can have better sense regarding the real-world application and how practical of the proposed algorithms.
2, Please clarify the challenges in theoretical analysis given existing methods.

---

> ### Author Rebuttal · Authors · 2024-08-06
>
> We thank the reviewer for the careful reading and thoughtful comments. We address the reviewer's questions in the following and will revise our paper accordingly. We hope the responses below address the reviewer's concerns.
>
>
> **Q1:** The lack of experiments on real-world dataset scenarios is a significant limitation. The authors should address how their proposed algorithms would perform and be applicable to practical problems in domains like recommender systems or healthcare. Without evaluations on real-world datasets, it is challenging to judge the practical utility and potential challenges of their methods.
>
> **A1:** Thank you for the insightful suggestion. We have performed experiments on a real-world dataset. Please refer to our Author Rebuttal.
>
> **Q2:** After given a tree-based method is used for private aggregation of the gradients of loss functions (Asi et al., 2021b), what's the challenge for the theoretical analysis of proposed algorithms given Asi et al., 2022b and Asi et al., 2023?
>
> **A2:** The novelty of our theoretical analysis in Theorem 1 is in addressing the unique challenges of a multi-client federated setting, which is not covered in previous works like Asi et al. (2021b, 2022b). While these prior works use a tree-based method for private aggregation of gradients in a single-player context, our approach involves multiple clients who add noise to their gradient estimates locally. This introduces new challenges in establishing the regret upper bound due to the aggregated effect of noisy updates from all clients.
>
> Specifically, in Fed-DP-OPE-Stoch, each client sends $\\{\langle c\_n,v\_{i,j,s} \rangle +\xi\_{i,n}\\}\_{n\in [d]}$, where $\xi\_{i,n}\sim \text{Lap}(\lambda\_{i,j,s})$, and $\lambda\_{i,j,s}= \frac{4\alpha2^j}{b\varepsilon}$. The server aggregates these to predict a new expert, i.e., $\bar{w}\_{j,s}=\underset{c\_n: 1\leq n\leq d}{\arg\min}\left[\frac{1}{m}\sum_{i=1}^m \big( \langle c\_n,v\_{i,j,s} \rangle + \xi\_{i,n} \big) \right]$.
>
> The key novelty and challenge are in handling the noisy estimates aggregated from multiple clients ($\frac{1}{m}\sum_{i=1}^m \big( \langle c_n,v_{i,j,s} \rangle + \xi_{i,n} \big)$) and analyzing its impact on the regret upper bound. To handle the impact of the average estimates $\frac{1}{m}\sum_{i=1}^m \langle c_n,v_{i,j,s} \rangle$, we show in Lemma 2 that every index of the $d$-dimensional vector $\nabla L\_{t}(x\_{i,p,k})-\bar{v}\_{p,k}$ is $O(\frac{1}{bm})$-sub-Gaussian by induction on the depth of vertex. This enables us to achieve $\sqrt{m}$-fold regret speed-up compared with Asi et al. (2022b). To deal with the challenge introduced by the aggregated noise $\frac{1}{m}\sum_{i=1}^m \xi_{i,n}$, we introduce Lemma 4, which quantifies the bound on the sum of $m$ IID Laplace random variables. This differentiates our analysis from prior single-client studies (Asi et al. 2021b, 2022b).
>
> Additionally, the federated setting poses challenges in reducing communication costs while maintaining privacy. Our Fed-DP-OPE-Stoch algorithm introduces a strategic communication protocol where communication is triggered only when the DP-FW subroutine, used for private gradient aggregation at each client, reaches a leaf vertex in the binary tree structure (Line 7-8 in Algorithm 1). This approach contributes to the overall reduction in communication costs, achieving logarithmic communication costs (Corollary 1). This aspect of Fed-DP-OPE-Stoch is another novel contribution, addressing the critical challenge of efficient communication in the federated setting. In comparison, Asi et al. (2021b, 2022b) focus on single-player settings and do not involve methods to control the communication cost.
>
> **Q3:** Please add at least one experiment with real-world dataset. So reader can have better sense regarding the real-world application and how practical of the proposed algorithms.
>
> **A3:** Thanks for the suggestion. We include it in our Author Rebuttal.
>
> **Q4:** Please clarify the challenges in theoretical analysis given existing methods.
>
> **A4:** We present the following challenges and contributions in the theoretical analysis.
>
> 1. Regret Upper Bound for Stochastic Adversaries: The challenges in the theoretical analysis of Fed-DP-OPE-Stoch and our contributions are detailed in **A2**.
>
> 2. Novel Lower Bounds for Oblivious Adversaries: We establish new lower bounds for federated OPE with oblivious adversaries, showing that collaboration among clients does not lead to speed-up in regret minimization (Theorem 2). The key challenge is formulating an instance of oblivious adversaries where the collaborative nature of federated learning does not result in the expected improvements in regret minimization. To address this, we propose a novel *policy reduction approach in FL*, representing a technical breakthrough (highlighted in the proof sketch of Theorem 2). Specifically, by defining an "average policy" among all clients against a uniform loss function generated by an oblivious adversary, we reduce the federated problem to a single-player one, showing the equivalence of per-client and single-player regret. To the best of our knowledge, our lower bounds represent the **first** of their kind for differentially private federated OPE problems.
>
> 3. Lower Bound for Oblivious Adversaries under Realizability Assumption: We establish a new lower bound in this setting (Theorem 3). Our analysis considers a specific oblivious adversary (detailed in the proof sketch of Theorem 3), which differentiates it from single-player scenarios (Asi et al., 2023). Our lower bound indicates that Fed-SVT is nearly optimal up to logarithmic factors (Remark 6).
>
> ---
>
> We thank the reviewer again for the helpful comments and suggestions for our work. If our response resolves your concerns to a satisfactory level, we kindly ask the reviewer to consider raising the rating of our work. Certainly, we are more than happy to address any further questions that you may have.

---

> ### Author Response · Authors · 2024-08-12
>
> Dear Reviewer NyWa,
>
> We've taken your initial feedback into careful consideration in our response. Could you please check whether our responses have properly addressed your concerns? If so, could you please kindly consider increasing your initial score accordingly? Certainly, we are more than happy to answer your further questions.
>
> Thank you for your time and effort in reviewing our work!
>
> Best Regards,
>
> Authors

---

> > ### Comment · Reviewer_NyWa · 2024-08-13
> >
> > Thank you for responding. I read the rebuttal and it addressed my second concern regarding novelty. I increased my score from 5 to 6.

---

> > > ### Author Response · Authors · 2024-08-13
> > >
> > > Thank you very much for your positive feedback! Your valuable input has helped us improve the quality of our work significantly. We really appreciate it.

---

### Official Review · Reviewer_H7Yw · 2024-07-27

**Soundness:** 3
**Presentation:** 3
**Contribution:** 3
**Rating:** 7
**Confidence:** 2

**Summary:**

The paper studies the problems of differentially private federated online prediction from experts against both stochastic adversaries and oblivious adversaries. The main contributions are three-fold. First, for stochastic adversaries, the paper proposes a differentially private mixture of experts algorithms and provide theoretical guarantees along with. Besides, for oblivious adversaries, the paper first shows a pessimistic result by providing a lower bound showing that federated learning cannot improve over the single machine learning in general. Furthermore, the paper shows that under special realizable case, however, federated learning can benefit from more machines. The paper proposes an algorithm that achieves that benefit with theoretical guarantees.

**Strengths:**

Both the theoretical analysis and algorithm design seems original and non-trivial. The technical contribution seems solid. The paper is also clear in its problem setup, algorithm description and theoretical justification. The problem solved by the paper is a concrete theoretical problem, and the paper does a good job in solving the problem overall.

**Weaknesses:**

My only concern of the paper is on the application side. While the problem studied by the paper makes perfect sense from a theoretical perspective, it does not seem very clear to me where it can have applications. With that being said, I think it is fine for authors to focus on a theoretical problem and leave its potential applications for future work.

**Questions:**

Would you please list some potential applications of the proposed algorithms? It will be better to have some specific applications for readers to keep in mind.

**Limitations:**

Like admitted by the authors and pointed out in the weakness section, the main limitation of the paper is the lack of real-world data experiments and applications. But given the theoretical nature of the paper, I do not think this limitation is unacceptable.

---

> ### Author Rebuttal · Authors · 2024-08-06
>
> We thank the reviewer for the careful reading and thoughtful comments. We address the reviewer's questions in the following and will revise our paper accordingly. We hope the responses below address the reviewer's concerns.
>
>
> **Q1:** My only concern of the paper is on the application side. While the problem studied by the paper makes perfect sense from a theoretical perspective, it does not seem very clear to me where it can have applications. With that being said, I think it is fine for authors to focus on a theoretical problem and leave its potential applications for future work. Would you please list some potential applications of the proposed algorithms? It will be better to have some specific applications for readers to keep in mind.
>
>
> **A1:** Thank you for the helpful comment. Differentially private federated online prediction has many important real-world applications. We provide three examples below, and will add the discussions in our revision.
>
> **Personalized Healthcare**: Consider a federated online prediction setting where patients' wearable devices collect and process health data locally, and the central server aggregates privacy-preserving updates from devices to provide health recommendations or alerts. DP federated online prediction can speed up the learning process and improve prediction accuracy without exposing individual users' health data, thus ensuring patient privacy.
>
>
>
> **Financial Fraud Detection**: DP federated online prediction can also enhance fraud detection systems across banking and financial services. Each client device (e.g. PC) locally analyzes transaction patterns and flags potential fraud without revealing sensitive transaction details to the central server. The server's role is to collect privacy-preserving updates from these clients to improve the global fraud detection model. This method ensures that the financial company can dynamically adapt to new fraudulent tactics, improving detection rates while safeguarding customers' financial privacy.
>
>
>
> **Personalized Recommender Systems**: Each client (e.g. smartphone) can personalize content recommendations by analyzing user interactions and preferences locally. The central server (e.g. company) aggregates privacy-preserving updates from all clients to refine the recommendation model. Thus, DP federated online prediction improves the whole recommender system performance while maintaining each client's privacy.
>
>
>
>
> **Q2:** Like admitted by the authors and pointed out in the weakness section, the main limitation of the paper is the lack of real-world data experiments and applications. But given the theoretical nature of the paper, I do not think this limitation is unacceptable.
>
>
> **A2:** Thank you for valuable comment. We have performed experiments on a real-world dataset. Please refer to our Author Rebuttal.
>
> -----
>
> We thank the reviewer again for the helpful comments and suggestions for our work. We are more than happy to address any further questions that you may have.

---

> ### Author Response · Authors · 2024-08-12
>
> Dear Reviewer H7Yw,
>
> We've taken your initial feedback into careful consideration in our response. Could you please check whether our responses have properly addressed your concerns? We are more than happy to answer your further questions.
>
> Thank you for your time and effort in reviewing our work!
>
> Best Regards,
>
> Authors

---

### Author Rebuttal · Authors · 2024-08-06

We thank all reviewers for their feedback, which has greatly improved our paper. We are glad that our work is recognized for studying "an important and unexplored research area" (Reviewer RFE1) and developing Fed-DP-OPE-Stoch with "novel techniques" (Reviewer hhm9) to handle stochastic adversaries, as well as a "novel proof technique" (Reviewer hhm9) for lower bounds in the classical oblivious case. Our Fed-SVT algorithm achieves near-optimal regret in a near-realizable case of oblivious adversaries, "demonstrating the quality of the proposed solution" (Reviewer NyWa). Our theoretical analysis and algorithm design are "original and non-trivial" (Reviewer H7Yw) and we "empirically illustrate the impact of the results with reproducible experiments" (Reviewer hhm9). Below, we address a common question the reviewers have.

**Could you perform experiments on real-world dataset scenarios? (Reviewers: H7Yw, NyWa, RFE1)**

We use the MovieLens-1M dataset [1] to evaluate the performances of Fed-SVT, comparing it with the single-player model Sparse-Vector [2]. We first compute the rating matrix of 6040 users to 18 movie genres (experts) $R = [r\_{u,g}] \in \mathbb{R}^{6040\times 18}$, and then calculate $L = [\max (0,r\_{u,g^\star}-r\_{u,g})]\in \mathbb{R}^{6040\times 18}$ where $g^\star = \arg \max\_g \left( \frac{1}{6040} \sum_{u=1}^{6040} r\_{u,g} \right)$. We generate the sequence of loss functions $\\{l\_u\\}\_{u\in [6040]}$ where $l\_u = L\_{u,:}$. In our experiments, we set $m=10$, $T = 604$, $\varepsilon = 10$, $\delta = 0$ and run 10 trials. In Fed-SVT, we experiment with communication intervals $N =1, 30, 50$, where communication cost scales in $O(mdT/N)$. The per-client cumulative regret as a function of $T$ is plotted in Figure 1 in our uploaded PDF file. Our results show that Fed-SVT significantly outperforms Sparse-Vector [2] with low communication costs (notably in the $N=50$ case). These results demonstrate the effectiveness of our algorithm in real-world applications. We will perform more experiments to thoroughly validate the performance of both Fed-SVT and Fed-DP-OPE-Stoch, and include the results in the next version of this paper.

References

[1] Harper F M, Konstan J A. The movielens datasets: History and context[J]. Acm transactions on interactive intelligent systems (tiis), 2015, 5(4): 1-19.

[2] Asi H, Feldman V, Koren T, et al. Near-optimal algorithms for private online optimization in the realizable regime[C]//International Conference on Machine Learning. PMLR, 2023: 1107-1120.

---

### Decision · Program_Chairs · 2024-09-25

**Decision:**

Accept (poster)

**Comment:**

This paper studies the problems of differentially private federated online prediction from experts against both stochastic adversaries and oblivious adversaries.

All the reviewers agree the current paper provide interesting theoretical results and that the authors have successfully addressed most of the reviewers' concerns. In light of this, I recommend acceptance and encourage the authors to include additional experimental results to further validate the proposed methods in the revision.